# RAGGED: Towards Informed Design of Retrieval Augmented Generation Systems

## Abstract

Retrieval-augmented generation (RAG) systems have shown promise in improving task performance by leveraging external context, but realizing their full potential depends on careful configuration. In this paper, we investigate how the choice of retriever and reader models, context length, and context quality impact RAG performance across different task types. Our findings reveal that while some readers consistently benefit from additional context, others degrade when exposed to irrelevant information, highlighting the need for tuning based on reader sensitivity to noise. Moreover, retriever improvements do not always translate into proportional gains in reader results, particularly in open-domain questions. However, in specialized tasks, even small improvements in retrieval can significantly boost reader results. These insights underscore the importance of optimizing RAG systems by aligning configurations with task complexity and domain-specific needs. [1]

## 1 Introduction

Retrieval-augmented generation (RAG) (Chen et al., 2017; Lewis et al., 2020) is a technique widely applied to enhance the performance of top-performing LMs on knowledge-intensive generation tasks like document-based question answering (Karpukhin et al., 2020). Given a question, the technique uses a *retriever* model to obtain relevant passages from a corpus. These passages are then inputted to a *reader* model as context for answering a given question.

Although using RAG supposedly helps LMs generate "more specific and factually accurate responses" (Lewis et al., 2020), we show that, in practice, achieving the greatest benefits from RAG requires careful configuration of all components in the RAG pipeline. Existing literature provides mixed, even contradictory, suggestions for configuring RAG. While some early works suggest that providing more retrieved passages results in strictly better outputs (Izacard & Grave, 2021), others find there is a limit to that phenomenon as model performance saturates after some number of contexts (Liu et al., 2023). In fact, some find that providing a select set of passages (Asai et al., 2022), sentences (Wang et al., 2023), or tokens (Berchansky et al., 2023) outperforms providing as many contexts as possible. Others find that reader model performance declines (Cuconasu et al., 2024; Jiang et al., 2024) as the number of contexts gets too large. The complexity of choosing the number of passages is only one aspect of RAG configuration among many that we cover in our analysis framework.

To provide more concrete suggestions of the *best practices* under various cases, we introduce an analysis framework, RAGGED, short for "retrieval augmented generation generalized evaluation device", to study RAG configurations on a suite of representative document-based question-answering (DBQA) tasks, including open-domain datasets that are single-hop and multi-hop questions (Kwiatkowski et al., 2019; Yang et al., 2018), and special-domain questions from the biomedical domain. We cover a broad range of models to ensure a comprehensive analysis: for retrievers, we incorporate both sparse and dense retrievers; for readers, we cover proprietary API models such as GPT (Brown et al., 2020) and Claude (Enis & Hopkins, 2024), as well as open-checkpoint models including Flan (Chung et al., 2022; Tay et al., 2023), LLaMa (Touvron et al., 2023b) families.

In this paper, we address the following key research questions (Figure 1):

**R1: When does RAG improve performance over the no-context baseline?**(§4) We explore whether RAG consistently enhances reader performance across different reader models and datasets. This

---

[1]Code and data for our RAGGED framework is available at https://anonymous.4open.science/r/ragged-05BD

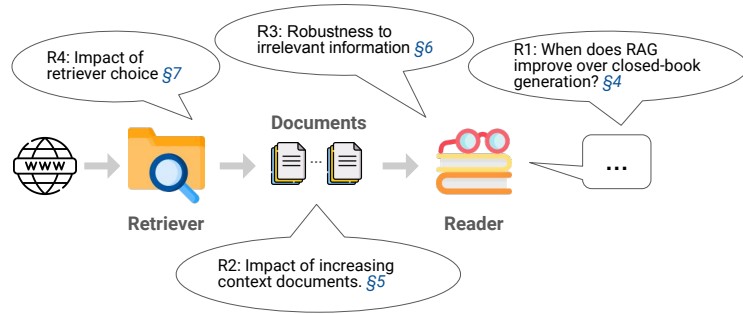

Figure 1: Roadmap of what our framework RAGGED analyses across the RAG pipeline.

analysis seeks to identify the specific scenarios — such as particular readers or question types — where RAG provides a clear advantage over closed-book generation, or whether its benefits are more situational.

**R2: How do reader models respond to an increasing number of context documents?**(§5) Building on R1, we investigate how reader performance is affected by the amount of context provided. Specifically, we examine whether adding more context passages improves model accuracy, leads to diminishing returns, or even degradation in performance due to too much noisy information (Figure 2).

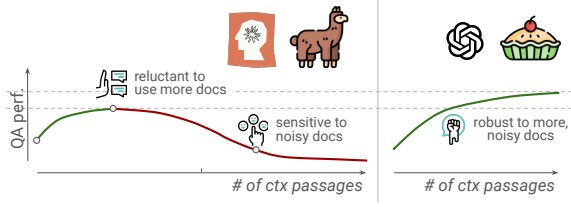

Figure 2: While some readers exhibit 'peak-then-decline' (left), others exhibit 'improve-then-plateau' behavior (right) with increasing number of contexts.

**R3: How robust are reader models to irrelevant information when relevant information is present or absent?**(§6) While R2 examines the effect of increasing context, it does not account for the quality of the context provided. In real-world scenarios, retrieved documents often contain both relevant and irrelevant information. We assess how reader models perform on data slices where relevant information is present and on slices where it is absent. This analysis is crucial for understanding model robustness to irrelevant information.

**R4: How does retriever choice impact reader performance across question types and domains?**(§7) To understand the impact of context quality from another perspective, we evaluate the effect of retriever model choice (i.e., sparse or dense) on reader performance across different question types (e.g., single-hop, multi-hop, and domain-specific questions). This investigation aims to identify the retriever-reader combinations that yield the best results depending on the task and domain.

In summary, our study provides actionable insights into when and how RAG can be effectively applied, offering guidance for configuring RAG systems to maximize their advantages. We introduce a reusable framework that can easily be used to analyze new retriever and reader models as they evolve. We release our full dataset and code, aiming to provide the community with a deeper understanding of the nuanced interplay between context quantity, quality, and model architecture in RAG systems.

## 2 THE RAGGED FRAMEWORK

In our analysis, we vary three key aspects:

1. **RAG system components:** We use three retrieval approaches: (1) BM25 (Robertson et al., 2009), a sparse retriever based on lexical matching; (2) ColBERT (Santhanam et al., 2021), a dense retriever using neural embeddings; and (3) Contriever (Izacard et al., 2022), an unsupervised dense retriever leveraging contrastive learning for efficient document retrieval. For readers, we examine closed-source models from the GPT and CLAUDE families, and open-source models from the FLAN, LLAMA2, and LLAMA3 families.

2. **Number of retrieved passages** ($k$)**:** We vary the number of retrieved passages from 1 to 50, with most insightful variations occurring before $k = 30$.

3. **Data slices based on retrieved passage quality:** Passage quality refers to the presence of "gold" passages, i.e., ground-truth passages, in the top-$k$ retrieved set.

We share the exact prompt in Appendix B.

## 3 EXPERIMENTAL SETUP

In this section, we describe the experimental setup, including the retriever and reader models, datasets, and evaluation metrics used to assess the performance of different RAG configurations.

### 3.1 RETRIEVER

We experiment with: (1) a sparse, lexical retriever and (2) a dense, semantic retriever.

**BM25**  BM25 (Robertson et al., 2009) is a probabilistic retrieval model that estimates passage relevance via term weighting and passage length normalization. It relies on term-matching and is supposed to be relatively proficient at identifying lexical similarity, especially in special domains.

**ColBERT**  One of the best-performing neural-based retrievers is ColBERT (Santhanam et al., 2021), i.e., contextualized late interaction over BERT. ColBERT uses contextualized embeddings instead of term-matching as in BM25, and is supposed to be relatively suited for identifying semantic similarities between queries and passages.

**Contriever**  Contriever (Izacard et al., 2022) is an unupservised, dense retrieval model. Contriever, like ColBERT, is also a dense retriever, but focuses more on overall semantic similarity instead of fine-grained matching by operating at the document level instead of the token level.

### 3.2 READER

We analyze closed-source models from the GPT and CLAUDE families and open-source models from the FLAN and LLAMA families.[2]

**FLAN**  The FLAN models are encoder-decoder models. We use the FLANT5-XXL (Chung et al., 2022) with 11B parameters and FLAN-UL2 (Tay et al., 2023) with 20B parameters, both with a context length of $2k$ tokens. FLANT5-XXL is an instruction-tuned variant of the T5 model (Raffel et al., 2023). FLAN-UL2 (Tay et al., 2023) is an upgraded T5-based model that is trained with Unifying Language Learning Paradigm, a pertaining process that uses a mixture-of-denoisers and mode switching to improve the model's adaptability to different scenarios.

**LLAMA**  We use 7B and 70B LLAMA2 models (Touvron et al., 2023a;b) and the 8B and 70B LLAMA3 models. The LLAMA2 models have a context length of 4k tokens while LLAMA3 models have double the context length to 8k tokens. The LLAMA3 models have three major differences: (1) They use grouped query attention, which groups query heads to understand similar information better, (2) Their vocabulary size is four times larger than that of LLAMA2, (3) They are trained on a seven times larger dataset than the LLAMA2 training corpus.

**GPT**  We mainly use GPT-3.5-turbo model (Brown et al., 2020). This model has a context length of 16k tokens and is a closed-source API model, so further details about model configurations are unknown. We also evaluate GPT-4O which has a context length of 128,000 tokens. However, due to the high cost, we only evaluate it on a small subset of experiments in Appendix K.

---

[2]While stronger LMs are available, we use ones with more affordable inference costs to allow for a large number of experiments over long contexts.

**CLAUDE** We use CLAUDE-3-HAIKU, which is Anthropic's fastest and most compact model (Enis & Hopkins, 2024). The context window of 200k tokens is the largest of all the models we compare in this paper, but the model size is unknown since the model is closed-source.

### 3.3 DATASETS

We adopt three DBQA datasets from various domains (Wikipedia, biomedical) and of various complexity (single-hop, multi-hop). More details are at Table 3 and Table 4.

**Natural Questions** We choose Natural Questions (NQ) (Kwiatkowski et al., 2019), a Wikipedia-based dataset, to examine how models perform on generic, open-domain, single-hop questions. NQ questions are actual user-search queries on Google. We adopt the KILT version (Petroni et al., 2021) of the dataset, which provides one short phrase answer and at least one gold passage for each question.

**HotpotQA** We choose HotpotQA (Yang et al., 2018), a multi-hop, Wikipedia-based dataset, to examine how effectively models can identify multiple signal passages and reason over them simultaneously. Each question requires reasoning over at least two passages to answer.

**BioASQ** We choose BioASQ's Task 11B (Krithara et al., 2023), a PubMed-based dataset, with biomedical questions to examine how models perform on special-domain questions. Our evaluation dataset is a compilation of the BioASQ Task 11B training and golden enriched set.

### 3.4 METRICS

**Retriever Metric** We evaluate retrieval performance using the **recall@k** metric, following Petroni et al. (2021). Recall@$k$ measures the fraction of ground-truth passages among the top-$k$ retrieved passages for a given query.

**Reader Metric** We use **unigram $F_1$**, which quantifies the overlap of unigrams in the reader output and gold answer(Petroni et al., 2021). For each query, we compute the $F_1$ score of the reader output against the list of gold answers and report the highest score. We also demonstrated a LLM-based, semantic metric using Kim et al. (2024) to evaluate correctness on a small subset and find the trends we observe using $F_1$ still hold Appendix H.

## 4 WHEN DOES RAG SURPASS THE NO-CONTEXT BASELINE?

Although RAG can potentially help ground LMs' generations in retrieved contexts, it is unclear how much these contexts help downstream performance, especially compared to a no-context baseline. While Lewis et al. (2020) achieve state-of-the-art results across several QA tasks by augmenting T5 model with a fixed $k$ number of documents, we find that the answer to the question of "When does RAG outperform no-context baseline" is more nuanced. To the best of our knowledge, we are the first to comprehensively explore this question across RAG configurations and datasets to find that while some readers always benefit from RAG, regardless of $k$, others benefit only if $k$ is large enough or small enough, then a few never do regardless of $k$ (Table 1). We first emphasize key observations in this section, then provide explanations in the latter sections.

**Closed-source models marginally improves with RAG** For instance, as shown in Table 1, GPT-3.5 generally performs better with RAG, regardless of the context quantity $k$. However, *its gains over its no-context baseline are marginal*. When averaged across values of $k$, GPT-3.5 only improves by 1.1 $F_1$ points, and its maximum gain is just 3.8 $F_1$ points (Table 5, Table 6). On NQ in particular, where GPT-3.5 already sets the no-context bar high at 52 $F_1$ points, it requires at least 5 passages to start benefitting from RAG with ColBERT. And in an extreme case, GPT-3.5, *when paired with BM25, always performs worse than its no-context baseline, regardless of $k$.*

CLAUDE-3-HAIKU struggles to benefit from RAG and is, in fact, harmed by using RAG for single-hop questions from both open-domain and special-domain (i.e., NQ, BioASQ). Even when it does benefit from RAG on HotpotQA, it only does so marginally with a gain of $< 4$ $F_1$ points.

| Dataset | ColBERT | BM25 | Dataset | ColBERT | BM25 |
|---|---|---|---|---|---|
| | **GPT-3.5-turbo** | | | **Claude-3-haiku** | |
| NQ | only for $k \geq 5$ | ✗ | NQ | ✗ | ✗ |
| HotpotQA | ✓ | ✓ | HotpotQA | only for $k \leq 2$ | ✓ |
| BioASQ | ✓ | ✓ | BioASQ | ✗ | ✗ |
| | **FlanT5** | | | **FlanUL2** | |
| NQ | ✓ | ✓ | NQ | ✓ | only for k > 3 |
| HotpotQA | ✓ | ✓ | HotpotQA | ✓ | ✓ |
| BioASQ | ✓ | ✓ | BioASQ | ✓ | ✓ |
| | **Llama2 7B** | | | **Llama2 70B** | |
| NQ | only for k < 10 | ✓ | NQ | only for k < 20 | ✓ |
| HotpotQA | ✓ | ✓ | HotpotQA | ✓ | ✓ |
| BioASQ | ✓ | ✓ | BioASQ | only for k < 20 | only for k < 20 |
| | **Llama3 8B** | | | **Llama3 70B** | |
| NQ | only for $k = 2$ | ✗ | NQ | ✗ | ✗ |
| HotpotQA | only for $k \leq 5$ | only for $k \leq 5$ | HotpotQA | only for $k \leq 5$ | only for $k \leq 5$ |
| BioASQ | only for $k \leq 2$ | only for $k \leq 2$ | BioASQ | only for $k = 1$ | ✗ |

Table 1: ✓ means the particular reader-retriever combination performs better than closed-book generation for all $k$'s. On the other hand, ✗ signifies that the particular reader-retriever combination consistently performs worse than closed-book generation, regardless of $k$. Otherwise, we describe the $k$-condition for which the retriever-reader combination performs better than closed-book generation.

**FLAN models greatly benefit from RAG**     FLANT5*'s ability to use context effectively allows it to significantly and consistently outperform closed-book generation across all datasets, retrievers, or $k$ number of contexts.* Across the datasets, FLANT5 with RAG achieves an average gain of 16 to 30 $F_1$ points, closely matching its optimal-$k$ gain (18 to 33 $F_1$ points), showing that it consistently benefits from using RAG. In fact, FLANT5 can use retrieved contexts so well that even though it ranks among the bottom three readers in terms of no-context performance, it ranks among the top-3 models in terms of optimal-$k$ RAG performance.

**LLAMA2 and LLAMA3 exhibit varied RAG abilities**     LLAMA2 models also often benefit from using RAG, but their performance is more sensitive to $k$. They often require a small enough $k$ to even benefit from using RAG, since a larger $k$ may introduce too much distracting information. Their sensitivity to $k$ is reflected in the large difference between their average gains (5 to 7 $F_1$ points) and optimal gains (8 to 12 $F_1$ points).

On the other hand, LLAMA3 models show minimal improvement, if any, from RAG. In many cases, they fail to outperform closed-book generation, regardless of $k$.

**Across question types and domains**     Among the datasets, performance on the multi-hop dataset consistently shows the highest gains from RAG in terms of average and optimal performance. One hypothesis is that language models often rely on memorized facts from pretraining for single-hop questions, making retrieval-augmented generation (RAG) less beneficial. In contrast, multi-hop questions require synthesizing information from multiple sources, which is less likely to be internalized during pretraining. Therefore, RAG could be more helpful for multi-hop tasks by providing disparate pieces of information that the model cannot generate from memory alone.

While Kandpal et al. (2023) do not explicitly compare open-domain (e.g., NQ) and specialized-domain (e.g., BioASQ) questions, they examine the effect of RAG relative to the number of relevant pretraining documents, finding that RAG outperforms closed-book generation, particularly for rare examples. Building on this, we hypothesize that open-domain questions may have more relevant pretraining documents while specialized-domain questions have fewer. Using BM25, we indeed find

that BioASQ enjoys a higher gain (4.31 $F_1$ points) than NQ (0.75 $F_1$ points). However, we find the opposite trend with ColBERT — NQ shows a larger gain (9.44 $F_1$) compared to BioASQ (7.26 $F_1$). This suggests that the question domain alone does not determine the relative optimal gain.

**Key Takeaways:** The benefits of RAG depend on the reader choice and their sensitivity to context quantity and quality. For some readers, the number of context passages is critical for achieving optimal performance. In such cases, a suboptimal RAG configuration can lead to worse performance than not using RAG. In the next section, we explore the influence of context quantity on RAG performance.

## 5 ARE MORE CONTEXTS ALWAYS BETTER?

In this section, we study how models perform with various amounts of retrieved passages in context. While Liu et al. (2023) report that reader performance saturates as $k$ increases, Cuconasu et al. (2024) and Jiang et al. (2024) observe performance degradation with increasing $k$. Although these findings appear contradictory, we argue that they are actually complementary, as each study focuses on a limited range of retrievers, readers, and datasets. Our experiments, which span a wider variety of retrievers, readers, and datasets, demonstrate that both saturation and degradation behaviors can occur *with the determining factor being the choice of reader model.*

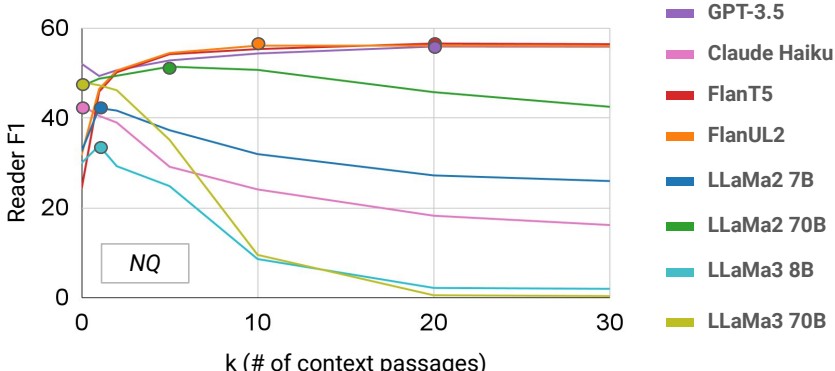

Figure 3: Reader performance on the NQ dataset as $k$, the number of contexts retrieved by ColBERT, varies. Colored circles indicate reader performance at the optimal $k^*$. We find similar trends apply across retrievers (e.g., BM25, ColBERT, and Contriever) and datasets (e.g., NQ, HotpotQA, and BioASQ) in Figure 11 and Figure 12.

We identify two distinct trends in reader performance in Figure 3.

**Improve-then-plateau** In the first, the FLAN models and GPT-3.5 steadily improve with increasing $k$ and then plateaus at around $k = 10$. *For such improve-then-plateau models, we recommend choosing a larger $k$ for maximizing downstream performance.*

**Peak-then-decline** In the second trend, models like the LLAMA models and CLAUDE-3-HAIKU peak early (at around $k < 5$) and then degrade with increasing $k$. *For such peak-then-decline models, a small $k$ is optimal*, as larger values introduce irrelevant or distracting information that can act as "noise" and harm performance.

Despite these trends, *a reader's response to increasing $k$ does not fully determine its overall ranking*. For instance, while the improve-then-plateau models achieve the top $F_1$ scores on NQ (55-56 $F_1$ score), peak-then-decline models like LLAMA2 70B still perform competitively (51 $F_1$ score). *However, there still is a distinguishing benefit of improve-then-plateau models, which is their robustness to changes in $k$*, making them less sensitive to variations once $k$ is sufficiently large. We have some hypotheses relating the readers' trends to their architectures and training details, but do not have access to the details of the closed-source models, so we only briefly discuss this according to the open-source models in Appendix E.

**Key Takeaways:** The choice of reader model is critical in optimizing RAG performance, as different models readers exhibit different sensitive to $k$. To efficiently explore RAG configurations, practitioners should start with a larger $k$ (e.g., $k \geq 20$) for improve-then-plateau models and a smaller $k$ (e.g., $k \leq 10$) for peak-then-decline models.

## 6 READER ROBUSTNESS TO NOISE IN RETRIEVED CONTEXTS

In this section, we take a deeper look at reader behaviors by studying how readers react to noise under contexts with (§6.1) and without gold passages (§6.2).

We analyze the reader model's performance on two slices of instances representing different qualities of retrieved contexts. In §6.1, we analyze the slice where the $k$ retrieved passages include at least one gold passage to mimic the scenario where sufficient context information is provided to answer the question. In this slice, we compare how the reader performs with (1) the top-k passages, (2) only the gold passages in the top-k passages (top-gold), and (3) no context. In §6.2, we analyze the slice of instances where none of the top-k retrieved passages are gold passages. This represents the scenario where the retrieved context is insufficient to answer the question.

### 6.1 WITH GOLD PASSAGES

For each model, we take the slice of examples with at least one gold passage in top-$k$ retrieval. We compare the top-$k$ performance with two baselines: (1) To study the adversarial effect of noisy context, we compare top-$k$ with *top-gold* performance where only the gold passages in top-k retrieval are included in context; and (2) to study the benefit of mixed quality documents, we compare top-$k$ and the *no-context* baseline where no documents are added to the context.

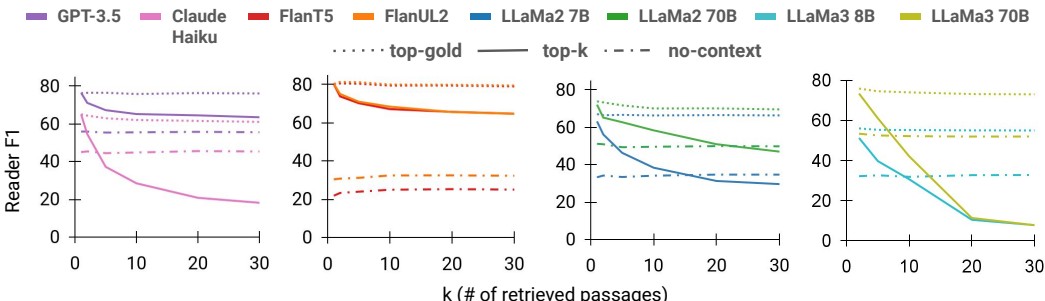

Figure 4: NQ results when sufficient information (at least one gold passage) is present in the top-k passages. 'Top-gold' refers to the context containing only the gold passages from the top-k passages.

**No-context is Not Always a Lower Bound for RAG, even with Context Signal** Reader models with only gold passages expectedly serve as an upper bound for their top-$k$ performance (Figure 4). However, it is notable that *the no-context performance does not always represent a lower bound for RAG*. Whether it is a lower bound depends on the reader's ability to filter out irrelevant information while leveraging helpful context information.

For NQ instances where gold passages are in the top-$k$ retrieved passages, GPT-3.5 and FLAN models consistently outperform their no-context baselines, effectively identifying and using relevant information. In contrast, models like CLAUDE-3-HAIKU and the LLAMA models struggle more with noise. CLAUDE-3-HAIKU and the LLAMA2 models fall below their no-context performance at $k \leq 5$ and LLAMA3 models do so at $k = 10$. This illustrates how *suboptimal RAG configurations can be not only less helpful but even damaging, even when sufficient information is present.*

**Multi-hop Signal Delays Performance Decline** In HotpotQA, the LLAMA2 models maintain performance above the no-context baseline longer than in NQ, with LLAMA2 7B dipping below at $k = 25$ instead of at $k = 15$ and LLAMA2 70B dropping below the no-context baseline at $k \geq 30$ instead of at $k = 25$ (Figure 6). Similarly, CLAUDE drops below the baseline at $k > 5$ instead of $k \leq 5$. This could suggest that *tasks requiring multiple signal passages provide more "anchor points" for the model, helping it withstand more noise.*

**Domain-specific Jargon Enables Easier Signal Extraction for Readers**  For BioASQ, *all readers' gaps between their top-pos and top-k performances are smaller than their gaps on open-domain datasets* (Figure 7), indicating better signal extraction likely due to the specialized domain jargon making relevant documents more distinct. We attribute the smaller gap primarily to the reader instead of the retriever since the retrieval quality for BioASQ is strictly worse than NQ (Table 8). Also of note is that CLAUDE-3-HAIKU and LLAMA3 70B still fall below their no-context baselines even with gold passages, showing that they struggle particularly with specialized domains. In these cases, the models often generate nonsensical outputs as $k$ increases.

## 6.2  WITHOUT GOLD PASSAGES

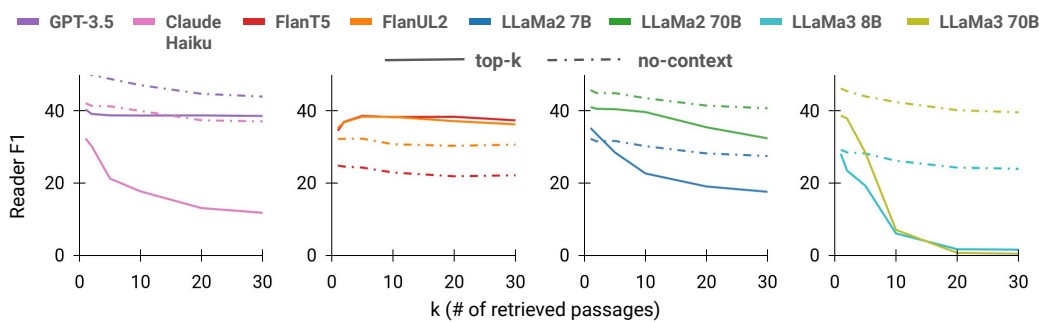

Figure 5: NQ results with no gold passages.

We conduct a similar analysis with examples retrieved with only non-gold passages. For NQ (Figure 5) and HotpotQA (Figure 8), most models perform worse with RAG than without. This is expected since these slices the models are prompted to rely on do not contain any signal (i.e., gold passages).

**FLAN Outperforms Baseline with Partially Relevant Non-gold Passages**  In contrast, FLAN *models consistently outperform their no-context baselines even with non-gold contexts*. One potential explanation is that the non-gold passages may still provide partially relevant information despite insufficient information. In particular, FLAN models seem better than other readers at processing the information from these non-gold passages. For example, on NQ with $k = 5$, the FLAN models achieve 20% accuracy when no gold paragraphs are retrieved but paragraphs from the gold Wikipedia pages are present. This is notably higher than the performances from LLAMA2 (8%) LLAMA3 models (4%), GPT-3.5 (10%), and CLAUDE-3-HAIKU (3%).

Another anomaly is that for BioASQ, GPT and LLAMA2 7B's top-$k$ performances both exceed their no-context baselines at select ranges of $k$ (Figure 9). This contrasts with how their top-$k$ performance is consistently worse than their no-context performance on the open-domain datasets, regardless of $k$. This suggests these models potentially have stronger guardrails for handling irrelevant information in specialized-domain questions.

**Key Takeaways:**  Reader performance in RAG systems depends heavily on handling noisy or irrelevant contexts, which is critical for real-world applications where retrieval is imperfect. Robust models are better suited for production systems with variable retrieval quality, ensuring stable performance. For models sensitive to irrelevant information, noise-filtering techniques are essential to maintain performance. These findings highlight the need to tailor RAG configurations by selecting models with strong noise-handling capabilities and adapting retrieval strategies appropriately.

## 7  IMPACT OF RETRIEVER CHOICE

We examine the impact of retriever choice on RAG performance by comparing BM25 and ColBERT across two metrics. First, we evaluate the **average difference**, which is the mean difference between $F_1$ scores when the reader is paired with top-$k$ documents from ColBERT v. BM25, averaged across $k = 1$ to 50. The **optimal $F_1$ difference** measures the difference in the optimal-$k$ $F_1$ score for

each reader when paired with ColBERT v. BM25. The first quantity describes how consistently ColBERT's downstream advantage is over BM25, and the latter describes ColBERT's advantage when both retrievers perform at their optimal $k$ (Table 2). While we mainly study the impact of retriever choice here, we also include reranker results in Appendix L and find the observed reader trends remain.

| Model | Average Difference (across $k$) | | | Difference in Optimal Performance | | |
|---|---|---|---|---|---|---|
| | NQ | HotpotQA | BioASQ | NQ | HotpotQA | BioASQ |
| GPT-3.5 | 8.6 | 2.0 | 1.1 | 6 | 1 | 0 |
| Claude Haiku | 3.9 | 4.0 | 2.4 | **12** | 3 | **6** |
| FlanT5 | 12.6 | **10.5** | **4.2** | 9 | 3 | 4 |
| FlanUL2 | **12.9** | 2.0 | 1.9 | 9 | 2 | 3 |
| LLaMa2 7B | 3.6 | 0.9 | -0.3 | 10 | 4 | 1 |
| LLaMa2 70B | 2.6 | 0.7 | -0.2 | 4 | 2 | 0 |
| LLaMa3 8B | -0.7 | -2.2 | 1.4 | 6 | 2 | 1 |
| LLaMa3 70B | -1.9 | -2.7 | 1.5 | 8 | **6** | 4 |
| **Average** | 5.2 | 1.9 | 1.5 | 8 | 2.9 | 2.4 |

Table 2: For each reader, the average difference and optimal difference in $F_1$ scores between ColBERT and BM25 are reported. (See the main text above for detailed definitions.)

**Better Retriever $\neq$ Better RAG Performance**    If a retriever has better recall@k for a specific $k$, it typically leads to better reader performance at that $k$. However, for less robust, peak-then-decline models (e.g., LLaMA 2 and 3), we observe that the reader F1 score with ColBERT can still be lower than with BM25 at larger $k$'s *even when ColBERT achieves higher recall@k than BM25 across all $k$'s*. This is reflected in the negative average reader F1 difference in Table 2.

One potential reason this discrepancy arises is that the content retrieved beyond the gold paragraphs plays a critical role, particularly at higher $k$'s. Neural retrievers like ColBERT may introduce more semantically complex or noisy content, which can overwhelm noise-sensitive readers. In contrast, BM25, as a lexical retriever, often provides simpler or less distracting context, aligning better with these readers' preferences despite retrieving fewer gold paragraphs overall.

**Large Retriever Gains only deliver Modest Gains for Open-domain Questions**    While ColBERT delivers substantial improvements in recall for open-domain questions (21.3 recall@k gain for NQ and 14.6 for HotpotQA) , the corresponding reader performance gains are more modest — 5.2 and 1.9 $F_1$ points, respectively. In fact, the ratio of reader gain to retriever recall improvement is quite low for HotpotQA (0.13) than NQ (0.24), indicating that a significant improvement in retriever may not yield an equally significant reader performance boost for open-domain questions.

**Small Retriever Gains yield Large Reader Gains for Specialized Domains**    In contrast, BioASQ's ratio of reader $F_1$ gain to retriever recall gain is much higher at 2.08. Although ColBERT's recall improvement over BM25 for BioASQ is small (0.7 vs. 14.6 for HotpotQA), the reader performance gains are comparable across both datasets. This suggests that *in specialized domains, even a small improvement in retriever performance can have an outsized impact on reader results.*

**Computational Trade-offs**    Given how 1) ColBERT only results in small optimal reader gains for HotpotQA and BioASQ and 2) BM25 is less computationally expensive to use, it may be tempting to claim BM25 is the obvious pick for RAG, computationally speaking. However, another important factor to consider is the difference in optimal $k$ — *the optimal $k$ with BM25 performance is 2 to 3 times that of the optimal $k$ for ColBERT* (Table 7). This means BM25's higher $k$ shifts the computational burden from the retriever to the reader, where the cost of inference is scaled with $k$.

**Key Takeaways:**    Using a neural retriever instead of a lexical one does not always lead to better reader performance, especially for less robust models and higher $k$'s. This highlights the importance of understanding the retriever-reader compatibility, and not just evaluating the retriever components independently and assuming aligned reader performance.

Nevertheless, we find that despite smaller retriever gains in specialized domains, we observe that the downstream reader performance is more significantly amplified. Even marginal improvements in retrieval can be highly valuable for domain-specific applications.

## 8 RELATED WORK

**Impact of Varying Number and Quality of Contexts**   When deciding the number of retrieved contexts to input into the reader model, existing findings provide seemingly contradicting findings. Izacard & Grave (2021) find that increasing the number of contexts provides strictly better results, while Liu et al. (2023) find the initial improvement is followed by performance saturation. On the other hand, some find that carefully selecting a subset of passages (Asai et al., 2022), sentences (Wang et al., 2023), or tokens (Berchansky et al., 2023) can outperform inputting maximal contexts. In fact, Cuconasu et al. (2024); Jiang et al. (2024) find that performance degrades with increasing $k$. Although existing findings seem contradictory, we think that they are actually complementary but restricted by their limited experiments. We experiment with a wider variety of configurations and datasets and arrive at a more nuanced conclusion: both the behavior of improve-then-plateau and peak-then-decline exist, and the primary deciding factor is the choice of reader.

**Domain Influence on Downstream Performance**   It is crucial to know when LMs benefit from including retrieved passages in context, especially for special domains, which are of particular interest in practice. Mallen et al. (2023) and Kandpal et al. (2023) both find that RAG helps for rare or long-tail knowledge. However, Mallen et al. (2023) find that retrieving contexts may be unnecessary and even detrimental when asking about common knowledge, while Kandpal et al. (2023) find using RAG generally improves model performance. From extensively experimenting with various RAG configurations and datasets, we provide a more nuanced view: performance gain on special domains is not necessarily always larger than the gain on open domains, and instead largely depends on the retriever-reader combination.

**Impact of Retriever Choice**   Lewis et al. (2020) show that dense retrievers like DPR outperform sparse retrievers (BM25) in open-domain tasks and help readers achieve better downstream performance on an open-domain dataset (i.e., NQ). Finardi et al. (2024) similarly find a positive correlation between retriever and reader performance, though their study is limited to a single, special-domain dataset. Both works demonstrate that dense retrievers lead to better reader outputs. In contrast, our extensive experiments show that better-performing retrievers do not necessarily lead to better downstream reader performance. We also comment on *how* important retriever choice — we find that although the ratio of the reader gain to retriever gain is positive for all datasets, it is much larger for specialized domains than for open-domain questions.

## 9 CONCLUSION

We propose RAGGED, a framework designed to assist researchers and practitioners in making informed decisions about designing RAG systems, focusing on three key aspects: the number of contexts, the reader model, and the retriever model.

Using the framework, we demonstrate that while retrieval-augmented generation (RAG) systems offer significant potential, their effectiveness depends on careful configuration. We find that some readers benefit from additional context, while others degrade when exposed to irrelevant information, underscoring the need for tuning based on reader behavior. Additionally, retriever improvements do not always lead to proportional gains in reader performance, particularly in open-domain tasks requiring multi-step reasoning. However, in specialized tasks, even minor improvements in retrieval can substantially boost reader performance.

These findings highlight the importance of task-specific RAG configurations and suggest that future research should focus on refining the interaction between retrievers and readers, improving model robustness to noisy contexts, and optimizing for domain-specific applications. We hope that researchers and practitioners can use our framework to unlock the full potential of RAG systems in various applications.

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

## A  READER IMPLEMENTATION DETAILS

We truncate the context to make sure the the rest of the prompt still fits within a reader's context limit. Specifically, when using FLANT5 and FLANUL2 readers, we use T5Tokenizer to truncate sequences to up to $2k$ tokens; when using LLAMA models, we apply the LlamaTokenizer and truncate sequences by $4k$ tokens for LLAMA2 and $8k$ for LLAMA3. For closed-source models, we spent around $300. Subsequently, we incorporate a concise question-and-answer format that segments the query using "Question:" and cues the model's response with "Answer:", ensuring precise and targeted answers.

For our reader decoding strategy, we used greedy decoding with a beam size of 1 and temperature of 1, selecting the most probable next word at each step without sampling. The output generation was configured to produce responses with 10 tokens. The experiments were conducted on NVIDIA A6000 GPUs, supported by an environment with 60GB RAM. The average response time was $\sim$1.1s per query when processing with a batch size of 50.

## B  PROMPT

For all experiments, we use the following prompt:

> *Instruction:* Give simple short one phrase answers for the questions based on the context
> *Context:* [passage$_1$, passage$_2$, $\cdots$, passage$_k$]
> *Question:* [the question of the current example]
> *Answer:*

## C  DATASET DETAILS

All corpus and datasets use English.

For NQ and HotpotQA datasets in the open domain, we use the Wikipedia paragraphs corpus provided by the KILT benchmark (Petroni et al., 2021). For BioASQ, we use the PubMed Annual Baseline Repository for 2023 (of Medicine, 2023), where each passage is either a title or an abstract of PubMed papers. Dataset sizes are in Table 4.

The Medline Corpus is from of Medicine (2023) provided by the National Library of Medicine.

| Corpus | # of par | # of doc | Avg # of doc |
|---|---|---|---|
| Wikipedia | 111M | 5M | 18.9 |
| Medline | 58M | 34M | 1.7 |

Table 3: Retrieval corpus information

For NQ and HotpotQA, we use KILT's dev set versions of the datasets, allowed under the MIT License (Petroni et al., 2021). For BioASQ (Krithara et al., 2023), we use Task 11B, distributed under CC BY 2.5 license.

| Dataset | # of Queries |
|---|---|
| NQ | 2837 |
| HotpotQA | 5600 |
| BioASQ | 3837 |

Table 4: Dataset information

## D  COMPARISON WITH NO-CONTEXT PERFORMANCE

We include additional reader results comparing ColBERT and BM25 at Table 5 and Table 6.

| Model | NQ | | HotpotQA | | BioASQ | |
|---|---|---|---|---|---|---|
| | **ColBERT** | **BM25** | **ColBERT** | **BM25** | **ColBERT** | **BM25** |
| GPT-3.5 | 1.1 | -8.0 | 7.7 | 5.1 | 8.9 | 7.4 |
| Claude Haiku | -15.7 | -22.5 | -6.5 | -10.2 | -21.7 | -25.9 |
| FlanT5 | **28.9** | **12.6** | **20.9** | 13.5 | **16.6** | **11.9** |
| FlanUL2 | 21.5 | 4.9 | 18.9 | **15.7** | 5.3 | 2.8 |
| LLaMa 2 7B | 1.4 | -4.5 | 10.4 | 8.2 | 6.4 | 5.9 |
| LLaMa 2 70B | -0.1 | -7.6 | 11.2 | 9.2 | 4.4 | 3.9 |
| LLaMa 3 8B | -13.3 | -14.9 | -6.7 | -5.7 | -12.1 | -14.9 |
| LLaMa 3 70B | -24.9 | -26.2 | -12.0 | -11.3 | -18.0 | -21.4 |
| **Average (per dataset)** | -0.1 | -8.3 | 5.5 | 3.06 | -1.3 | -3.8 |

Table 5: The average difference between the $F_1$ score of RAG with $k$ passages from ColBERT or BM25 and the $F_1$ score of no-context generation, calculated across $k$ values from 1 to 50 for each dataset. Each value represents the difference between the $F_1$ score of the reader+retriever combination and the $F_1$ score of the reader alone (without RAG or context).

| Model | NQ | | HotpotQA | | BioASQ | |
|---|---|---|---|---|---|---|
| | **ColBERT** | **BM25** | **ColBERT** | **BM25** | **ColBERT** | **BM25** |
| GPT-3.5 | 3.8 | -3.1 | 8.8 | 7.3 | 10.9 | 10.6 |
| Claude Haiku | -2.4 | -14.9 | 3.9 | 0.7 | -1.7 | -8.2 |
| FlanT5 | **32.5** | **22.6** | **23.5** | **20.4** | **18.0** | **13.0** |
| FlanUL2 | 24.4 | 14.8 | 22.0 | 19.7 | 7.1 | 3.4 |
| LLaMa 2 7B | 9.7 | -0.5 | 15.2 | 11.0 | 8.5 | 6.9 |
| LLaMa 2 70B | 4.3 | -0.3 | 14.0 | 11.4 | 7.3 | 6.8 |
| LLaMa 3 8B | 3.9 | -3.0 | 11.2 | 8.4 | 3.6 | 1.7 |
| LLaMa 3 70B | -0.7 | -9.6 | 14.9 | 8.1 | 4.4 | 0.3 |
| **Average (per dataset)** | 9.44 | 0.8 | 14.2 | 10.9 | 7.3 | 4.3 |

Table 6: The difference between the $F_1$ score of RAG optimal $k^*$ from ColBERT or BM25 and the $F_1$ score of no-context generation. Each value represents the difference between the $F_1$ score of the reader+retriever combination at optimal $k^*$ and the $F_1$ score of the reader alone (without RAG or context).

# E  RELATING READER TRENDS TO READER ARCHITECTURES AND TRAINING DETAILS

There are two primary types of readers observed in our experiments:

- Peak-then-Decline Behavior: Models including those from the LLaMa and Claude families show sensitivity to noisy documents, leading to performance degradation as the number of retrieved passages (k) increases beyond a certain point.
- Improve-then-Plateau Behavior: Models including those from the GPT and FLAN families are more robust to noise, continuing to benefit from additional context until performance plateaus.

Since we do not have access to the details of the closed-source models, we will focus on providing hypotheses according to the open-source model (LLaMa belonging to the peak-then-decline behavior and the FLAN models belonging to the improve-then-plateau family).

On one hand, FLAN, an improve-then-plateau model family, incorporates additional strategies explicitly designed to handle noisy or diverse contexts. It employs denoising strategies, such as a mixture-of-denoisers, during training to improve its robustness to irrelevant or noisy contexts. These enhancements enable it to filter out noise more effectively.

On the other hand, LLAMA 's training predominantly relies on next-token prediction with limited exposure to noisy or retrieval-specific scenarios, making it sensitive to noise at higher k.

We also note that there are some model architecture features that alone do not determine reader behavior:

- Context window size: Models with longer context limits like LLAMA 2 (4k tokens) don't necessarily process a larger number of contexts better than models with smaller context limits like FLAN (2k tokens).
- Encoder-decoder v. decoder: LLAMA is a decoder-only model that displays peak-then-decline behavior, but GPT models are also decoder-only and instead display improve-then-plateau behavior.

## F    SLICE ANALYSIS ON OTHER DATASETS

We include *with*-gold-passages results for HotpotQA at Figure 6 and for BioASQ at Figure 7.

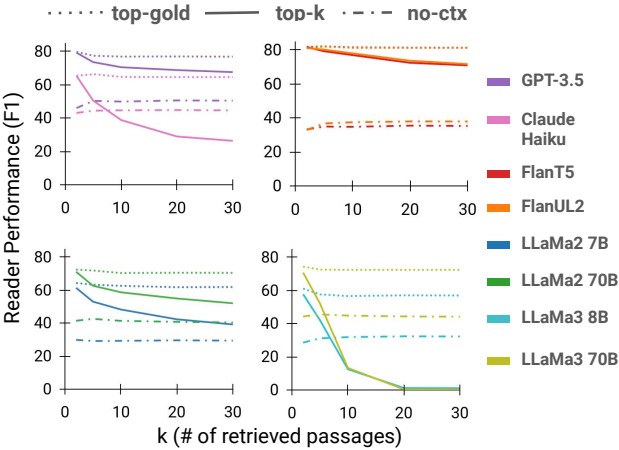

Figure 6: HotpotQA results when there is sufficient information (all gold passages) included in the top-k passages to answer the question. For multi-hop questions, we select examples retrieved with all gold passages within the top-$k$ passages since all passages are necessary to answer the question.

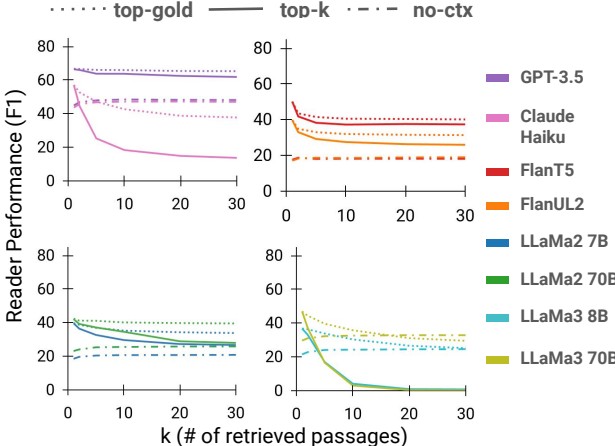

Figure 7: BioASQ results when there is sufficient information (at least one gold passage) included in the top-k passages to answer the question.

We include *without*-gold-passages results for HotpotQA at Figure 8 and for BioASQ at Figure 9.

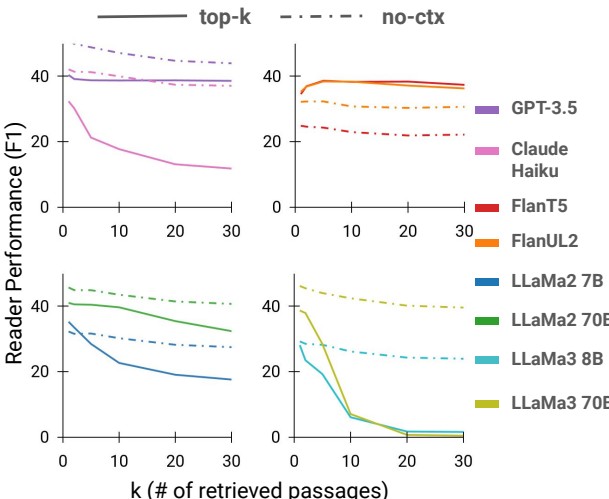

Figure 8: HotpotQA results when there are no gold passages included in the top-k passages to answer the question.

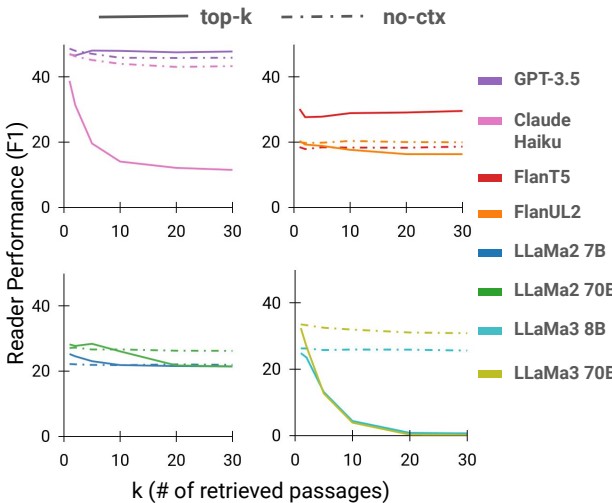

Figure 9: BioASQ results when there are no gold passages included in the top-k passages to answer the question.

## G  COMPARING OPTIMAL K VALUES

We include the optimal $k$ for ColBERT and BM25 in Table 7.

| Model | NQ | | HotpotQA | | BioASQ | | Average (per reader) | |
|---|---|---|---|---|---|---|---|---|
| | BM25 | ColBERT | BM25 | ColBERT | BM25 | ColBERT | BM25 | ColBERT |
| GPT-3.5 | 50 | 20 | 50 | 20 | 20 | 20 | 40 | 20 |
| CLAUDE Haiku | 1 | 1 | 1 | 1 | 1 | 1 | 1 | 1 |
| FlanT5 | 50 | 20 | 10 | 10 | 50 | 1 | 36.67 | 10.33 |
| FlanUL2 | 50 | 10 | 20 | 10 | 2 | 1 | 24 | 7 |
| LLAMA 2 7B | 1 | 1 | 2 | 2 | 2 | 1 | 1.67 | 1.33 |
| LLAMA 2 70B | 10 | 5 | 10 | 2 | 5 | 5 | 8.33 | 4 |
| LLAMA 3 8B | 1 | 1 | 1 | 1 | 1 | 1 | 1 | 1 |
| LLAMA 3 70B | 1 | 1 | 1 | 1 | 1 | 1 | 1 | 1 |
| Average (per dataset) | 20.5 | 7.38 | 11.88 | 5.88 | 10.25 | 3.88 | 14.21 | 5.71 |

Table 7: Optimal $k^*$ for BM25 and ColBERT (NQ, HotpotQA, and BioASQ).

## H  LLM-BASED EVALUATION

While we chose F1 for its simplicity and alignment with prior work, we agree that it may not fully reflect nuanced semantic equivalence. To address this, we ran an LLM-based evaluation of the models for the NQ dataset using Prometheus (Kim et al., 2024), specifically the Prometheus-7b-v2.0 model. We find that the conclusions about reader trends do not change: the same reader trends apply to the same models (peak-then-decline v. improve-then-plateau). We use Prometheus-7b-v2.0 to evaluate the correctness of the generated answer against the gold answer on a 5-point scale, where 1 is the least correct and 5 is the most correct Figure 10.

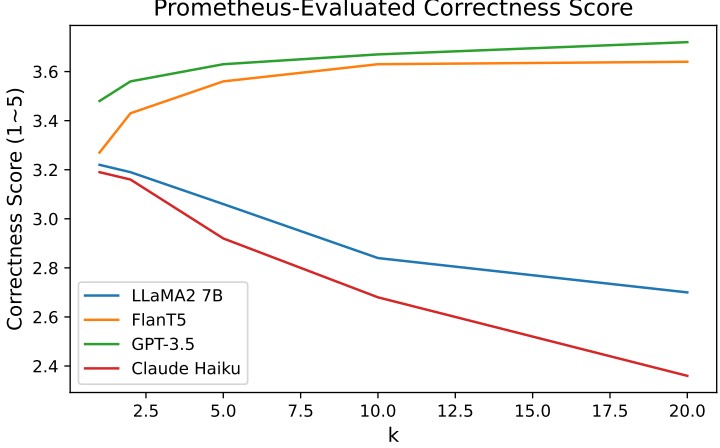

Figure 10: Reader Performance on NQ dataset as evaluated by Prometheus on a 5-point scale where 1 is the least correct and 5 the the most correct.

## I  COMPARING READER TRENDS WHEN USING COLBERT V. BM25

We include the top-k performance for ColBERT, BM25 Figure 11.

## J  COMPARING NEURAL RETRIEVERS

We compare the top-$k$ performance of ColBERT and Contriever at Figure 12.

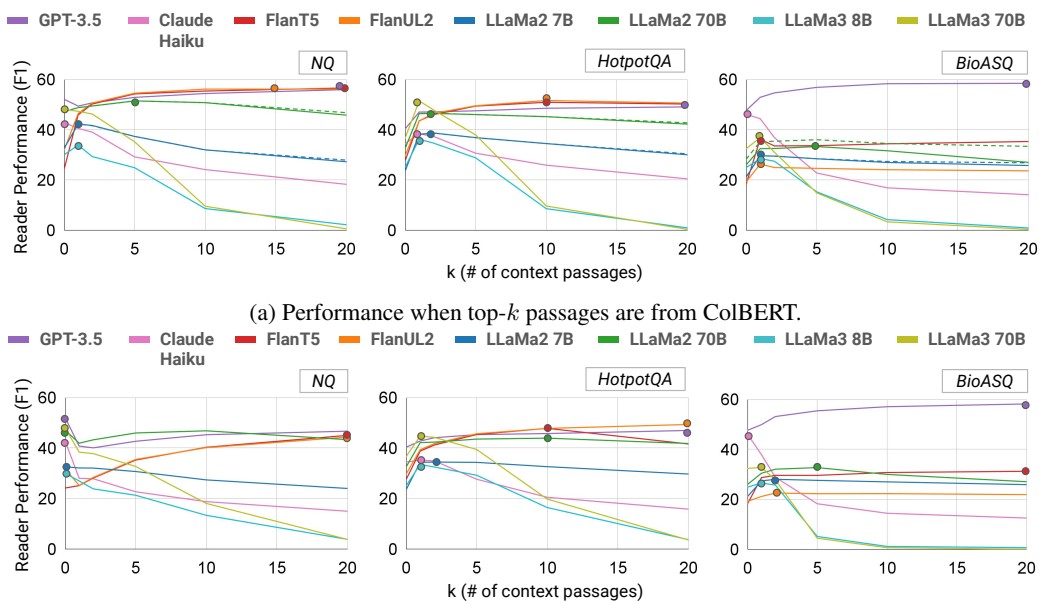

(a) Performance when top-$k$ passages are from ColBERT.

(b) Performance when top-$k$ passages are from BM25.

Figure 11: Top-k performance on NQ, HotpotQA, and BioASQ. Colored circles mark the reader performance at optimal $k^*$.

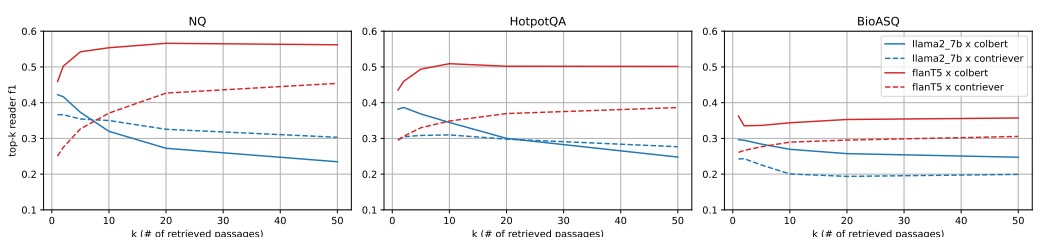

Figure 12: Example of how reader response to increasing context applies across neural retrievers (e.g., ColBERT and Contriever) and datasets. We choose one reader model from each trend for demonstration — LLaMa2 7B for peak-then-decline and FlanT5 for improve-then-plateau.

## K  COMPARING GPT-3.5 AND GPT-4O

We compare how GPT-3.5 and GPT-4O perform, and find that they both display the same reader trend of improve-then-plateau, with the main difference being GPT-4O's reader performance is shifted up (Figure 13).

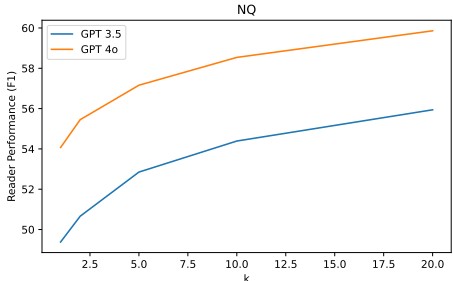

Figure 13: Comparison of GPT-3.5 and GPT-4O performance on NQ.

## L  EFFECT OF RERANKER

We use bge-reranker-large, which is currently the most downloaded reranker on Hugging Face (Xiao et al., 2024). We demonstrate the effect of reranker on ColBERT, the better-performing retriever in our experiments, and choose one open-source model from each reader robustness type (LLAMA 2 7b from peak-then-decline and FLAN T5 from improve-then-plateau). Of the top 50 documents from ColBERT, we apply the reranker to reassign scores, then get the new top-$k$ documents to feed to the readers. The result is that each of these models still displays the same reader trends, except the scores are shifted up. We present the F1 scores across k's in Figure 14

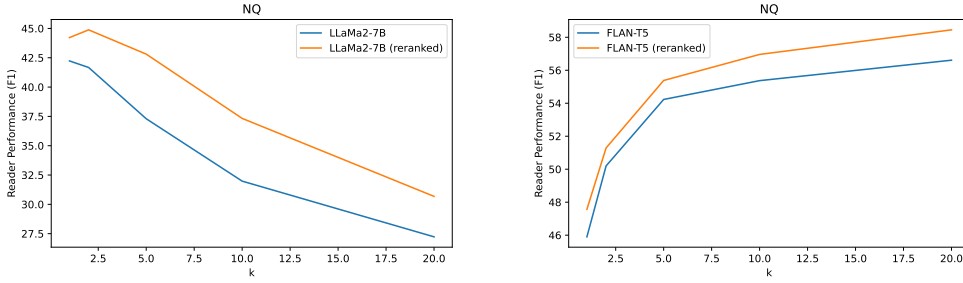

Figure 14: Comparison of reader performance with and without bge-large reranker on the NQ dataset.

# M RETRIEVER PERFORMANCE

We include the retriever performance at select $k$'s at Table 8.

| Retriever | Recall@k | | | | | |
|---|---|---|---|---|---|---|
| | 1 | 2 | 5 | 10 | 20 | 50 |
| *NQ* | | | | | | |
| BM25 | 2.7 | 4.4 | 8.0 | 11.5 | 16.3 | 22.8 |
| | 10.3 | 16.3 | 27.8 | 36.8 | 47.7 | 53.2 |
| ColBERT | 12.3 | 18.0 | 25.7 | 32.1 | 38.1 | 41.8 |
| | 27.2 | 38.8 | 54.4 | 65.0 | 72.9 | 77.2 |
| Contriever | 4.65 | 6.91 | 11.14 | 15.17 | 20.19 | 28.46 |
| | 24.0 | 32.3 | 44.9 | 53.2 | 62.1 | 72.0 |
| *HotpotQA* | | | | | | |
| BM25 | 19.1 | 25.9 | 34.6 | 41.1 | 46.8 | 54.2 |
| | 23.3 | 31.2 | 42.7 | 52.1 | 59.1 | 62.8 |
| ColBERT | 31.1 | 40.1 | 49.9 | 56.2 | 61.9 | 64.9 |
| | 34.2 | 44.7 | 56.3 | 63.6 | 69.9 | 73.1 |
| Contriever | 2.35 | 4.44 | 8.14 | 11.75 | 15.46 | 20.79 |
| | 22.39 | 29.54 | 39.39 | 45.71 | 51.51 | 59.08 |
| *BioASQ* | | | | | | |
| BM25 | 8.8 | 12.9 | 19.6 | 25.8 | 33.3 | 37.8 |
| | 12.4 | 16.4 | 23.9 | 30.6 | 38.7 | 43.6 |
| ColBERT | 8.8 | 13.5 | 20.7 | 27.1 | 34.3 | 38.6 |
| | 14.2 | 18.2 | 25.6 | 32.2 | 39.8 | 44.2 |
| Contriever | 3.82 | 5.87 | 9.55 | 12.95 | 17.48 | 24.58 |
| | 7.91 | 10.55 | 15.36 | 19.61 | 24.89 | 33.03 |

Table 8: Retriever performance (recall@k). For the Wikipedia-based dataset, the top row indicates recall@k at the retrieval unit of Wikipedia paragraph and the bottom row for the unit of Wikipedia page. For BioASQ, the top row indicates recall@k at the unit of title or abstract of a PubMed article and the bottom row at the unit of the article itself.

