# OpenReview forum: "RAGGED: Towards Informed Design of Retrieval Augmented Generation Systems"
_ICLR.cc/2025/Conference — Submitted to ICLR 2025_

### Official Review · Reviewer_e7Uq · 2024-11-03

**Soundness:** 2
**Presentation:** 3
**Contribution:** 2
**Rating:** 6
**Confidence:** 4

**Summary:**

The paper introduces RAGGED, a framework for analyzing and evaluating RAG systems. While the work presents a comprehensive analysis framework, its contributions mainly focus on investigating how different components and configurations affect RAG performance across various tasks and domains.
The main contributions include:
- A systematic analysis framework examining RAG components,
- Empirical findings on RAG performance characteristics,
- Investigation of retriever-reader relationships,
- Domain-specific insights for RAG applications.

**Strengths:**

- The study examines multiple dimensions of RAG systems with a systematic approach, using diverse models and configurations. This provides valuable empirical evidence about component interactions.
- The authors provide their code and framework publicly, with well-documented experimental setups and clear implementation details.
- The work challenges some common assumptions about RAG systems, particularly regarding the relationship between retriever and reader performance.

**Weaknesses:**

- While the paper presents extensive empirical findings about RAG system performance, it lacks a theoretical framework that could explain the underlying patterns and relationships between reader model's architectures (e.g., parameter count, token limits) and their performance characteristics, which could have enabled broader generalizations about unexplored model families and helped predict optimal configurations based on architectural features rather than just empirical observations.
- The evaluation focuses narrowly on question-answering tasks with relatively small datasets (2837 queries for NQ, 5600 for HotpotQA, 3837 for BioASQ). This raises questions about the generalizability of findings to other applications/domains.
- The paper could benefit from more rigorous comparison with existing RAG analysis frameworks or methodologies to better establish its unique contribution.

**Questions:**

- Could the authors analyze how model characteristics (e.g., parameter count, architecture type, pre-training approach, token limits) correlate with RAG performance patterns? Have you observed any clustering of performance patterns based on model families or architectures that could suggest theoretical principles for predicting RAG behavior? Such analysis might help generalize findings to unstudied models and provide theoretical grounding for the empirical observations.
- The significantly different behavior observed in specialized domains (e.g., BioASQ) vs. open domain is intriguing. Could you elaborate on what characteristics of specialized domains lead to these differences? Have you analyzed if reader models pre-trained on domain-specific data show different RAG behavior patterns?
- Could you provide insights into how to predict whether a domain would benefit more from dense vs. sparse retrievers?

---

> ### Author Response · Authors · 2024-11-22
> **Response to Reviewer e7Uq**
>
> Thank you for your review and for recognizing the systematic approach of our framework and the value of our empirical insights. We appreciate your acknowledgment of our contributions to understanding retriever-reader interactions and our commitment to reproducibility.
>
> __W1/Q1. Relation between reader’s architectural features and performance:__
> We agree that deeper insights into how model characteristics relate to model robustness could enhance the impact of our work. Per your suggestion, we discuss this in more detail in [this general comment thread](https://openreview.net/forum?id=KDXj60FpJr&noteId=ey7JaljKeE). Please refer to it for a comprehensive response, and feel free to follow up with any additional questions. We plan to include this in the revised paper.
>
> __W2. Size of datasets__
> Thank you for your feedback on the dataset sizes. While the datasets used (NQ, HotpotQA, and BioASQ) may appear small, they are widely used and accepted benchmarks in the research community. This ensures comparability with prior work and alignment with established evaluation standards. In addition to their popularity, we also choose these datasets so that they are complementary in their task coverage: NQ represents single-hop open-domain QA and serves as a standard dataset for general-domain retrieval evaluation, BioASQ provides a specialized-domain perspective, and HotPotQA adds diversity by evaluating multi-hop reasoning.
>
> __W3. Comparison with existing RAG analysis frameworks__
>
> While there does exist a popular RAG evaluation framework [1], their paper does not actually run extensive experiments using their framework and deriving insights from it. Instead, their contribution is primarily their new LLM-based evaluation metrics, such as faithfulness, answer relevance, and context relevance.
>
> __Q2: Specialized Domain behavior__
>
> Specialized domains (e.g., BioASQ) show higher average reader gain relative to average retriever gain compared to open domains. This is likely because readers rely more heavily on correct context information due to limited exposure to specialized knowledge during pretraining. Thus, even minor improvements in retriever performance can significantly amplify reader performance in these domains.
>
> __Q3: Dense vs. Sparse Retrievers__
>
> From what we have observed, in general domains (e.g., NQ), dense retrievers like ColBERT outperform sparse retrievers like BM25, providing a strong starting point.
>
> In specialized domains (e.g., BioASQ), ColBERT and BM25 perform similarly in retrieval metrics (e.g., recall@k). However, the reader gain due to retrieval improvement is more amplified in specialized domains, making it essential to test both retrievers and focus on even small retrieval improvements.
>
> Then, there is the question of what is considered a specialized domain to a reader model. If pretraining document access is available, a straightforward measure could be the document frequency of domain-specific content in the corpus. Domains with lower representation in pretraining data are likely to benefit more from accurate retrieval, regardless of retriever type.
>
> [1] ] S. Es, J. James, L. Espinosa-Anke, and S. Schockaert, “Ragas: Automated evaluation of retrieval augmented generation,” arXiv preprint arXiv:2309.15217, 2023.

---

> > ### Author Response · Authors · 2024-11-25
> > **Response to Reviewer e7Uq (Continued)**
> >
> > Dear reviewer, thank you again for your detailed and thoughtful feedback. We have tried to address your points by adding further analysis and clarifications. We were wondering if our responses address your concerns, or if there are any remaining points where further clarification would be helpful. We’d be happy to provide additional insights if needed. Thank you for your time and thoughtful consideration!

---

> ### Comment · Area_Chair_9EME · 2024-11-26
> **Reminder: Rebuttal Deadline for ICLR 2025**
>
> Dear Reviewer e7Uq,
>
> As the rebuttal deadline approaches, please kindly check the papers' discussion threads and respond to the authors' rebuttals. If you haven't had a chance to respond yet, I’d greatly appreciate your input soon. Your insights are invaluable to the authors and the review process.
>
> Thank you for your effort and support!
>
> Best regards,
>
> Area chair

---

### Official Review · Reviewer_F1ot · 2024-11-03

**Soundness:** 2
**Presentation:** 3
**Contribution:** 2
**Rating:** 3
**Confidence:** 5

**Summary:**

This paper proposes RAGGED, aiming to study the impact of different readers, retrievers, and the number of retrieved documents. It reveals the different response patterns of the reader model to contexts and provides suggestions for deploying an RAG system.

**Strengths:**

This paper proposes the RAGGED framework with the comprehensive and systematic study of multiple key aspects of RAG configuration. It constructs a complete assessment system through four clear research questions (R1-R4).

**Weaknesses:**

# Limitations in Methodological Approach

### Retrieval Architecture Constraints

- Limited Exploration of Modern Retrieval Methods
  - The study did not incorporate state-of-the-art retrievers (e.g., BGE) and reranking mechanisms
  - Contemporary search engines and their potential impact were not evaluated
- Generalizability Concerns
  - The conclusion that "better searchers ≠ better RAG performance" may require reconsideration in light of recent empirical evidence using modern retrieval systems

### Architectural Paradigm Limitations

- Restricted RAG Implementation Scope
  - The investigation did not encompass emerging RAG variants (e.g., Self-RAG) and recent Large Language Model paradigms (e.g., OpenAI O1)
  - The impact of prompt engineering variations was not systematically explored (e.g., CoT and Self-Ask)
- Retrieval Source Considerations
  - Limited investigation into how different knowledge source types affect system performance

### Dataset and Task Coverage

- The study's conclusions are constrained by the limited variety of datasets and task types examined (only 3 datasets)
- A broader spectrum of use cases would strengthen the generalizability of the findings



# Limitations in Experimental Depth and Analysis

## Experimental Design Constraints

### Variable Control Inadequacies

- Noise Distribution Effects: The impact of noise location within documents remains unexplored
- Document Structure Variables
  - The influence of paragraph length variation was not systematically analyzed
  - Potential correlations between document structure and retrieval performance were not examined

### Analytical Depth Limitations

- Lack of analysis of the reasons why "Some retriever-reader combinations consistently benefit from RAG, while others only sometimes do for certain k’s, and a few never do regardless of k"
- The analysis lacks depth, only revealing obvious, common, and widely known conclusions without examining the underlying causes. I suggest analyzing the conclusions mentioned in the article from the perspective of how LLMs handle conflicts and integration between their internal knowledge and external information.

## Practical Implementation Constraints

- Configuration Guidelines
  - Lack of concrete, actionable implementation recommendations
  - Modern RAG systems include many other components, such as rerankers, scoring mechanisms, query routing, and query rewriting. **Your work appears to be limited to performing grid search on just a few hyperparameters.**
- The applicability/generalizability to other tasks cannot be guaranteed.

**Questions:**

- Q1: Why weren't modern retrievers like BGE and search engines considered? Would this affect the reliability of the main conclusions?
- Q2: How was the strategy for selecting noise paragraphs determined? Does this choice affect the validity of the experiments?
- Q3: Does the conclusion "better retrievers ≠ better RAG performance" still hold after incorporating rerankers and more modern retrievers?
- Q4: Do the improve-then-plateau and peak-then-decline behavior patterns still exist with different RAG and LLM paradigms?
- Q5: Why was unigram F1 chosen instead of other potentially more suitable evaluation metrics? The paper mentions that “many LMs respond correctly semantically but use inexact or more verbose wording.” which can be avoided by extracting the answer using the LLM. The effect of the Unigram-F1 assessment is not comprehensive enough.
- Q6: What are the criteria for selecting noise paragraphs in Section 6?
- Q7: Was the impact of different retrieval passage lengths considered?
- Q8: Can you explain the underlying reasons why different models respond differently to RAG improvements?
- Q9: How do the quantity and scope of parametric knowledge correlate with performance variations across different LLMs? Additionally, could the authors explore the dynamics of knowledge conflicts and integration between an LLM's internal (learned during pre-training) and external (retrieved during inference) knowledge bases?
- Q10: The paper mentions providing "To provide more concrete suggestions of the best practices under various cases, we introduce an analysis framework, RAGGED," What are some efficient approaches to determine the optimal RAG configuration when working with new datasets or models? Specifically, how can practitioners rapidly identify the most effective combination of retrieval methods, LLMs, $k$, and other key parameters?

---

> ### Author Response · Authors · 2024-11-22
> **Response to Reviewer F1ot**
>
> Thank you for your review and for highlighting the systematic and comprehensive nature of the RAGGED framework. We are glad you found our use of clear research questions (R1-R4) effective in presenting our findings.
>
> __W1: Limited Retrieval Sources/Corpora__
>
> The open-domain questions use the Wikipedia domain as the corpus, and the BioASQ uses PubMed articles as the corpus. We are limited to these corpora due to the annotations available in the datasets: for each question, we need to be able to trace which documents are gold to evaluate retriever performance and to provide more detailed context signal-noise analysis.
>
> __W2: Finding Novelty__
>
> Thank you for your question regarding the novelty of the findings. We address this in this [general comment thread](https://openreview.net/forum?id=KDXj60FpJr&noteId=w3Xrhj82h3). Please refer to it for a comprehensive response, and feel free to follow up with any additional questions.
>
> __Q1: Modern Retrieval Methods__
>
> We appreciate your suggestion and have run experiments using bge-reranker-large, which is currently the most downloaded reranker on huggingface. Due to the limited rebuttal time frame, we select a subset of models and datasets to demonstrate the effect. We choose ColBERT, the better-performing retriever, and we choose one open-source model from each reader robustness type (LLaMa2 7b from peak-then-decline and FLAN T5 from improve-then-plateau).
> Of the top 50 documents from ColBERT, we apply the reranker to reassign scores, then get the new top-k documents to feed to the readers. The result is that each of these models still displays the same reader trends, except the scores are shifted up. We present the F1 scores across k’s.
>
>  | Model                | 1     | 2     | 5     | 10    | 20    |
> |----------------------|-------|-------|-------|-------|-------|
> | LLaMA2 7B           | 42.23 | 41.67 | 37.30 | 31.98 | 27.23 |
> | LLaMA2 7B reranked  | 44.22 | 44.88 | 42.81 | 37.33 | 30.68 |
>
> | Model            | 1     | 2     | 5     | 10    | 20    |
> |------------------|-------|-------|-------|-------|-------|
> | FLAN T5           | 45.90 | 50.20 | 54.23 | 55.37 | 56.61 |
> | FLAN T5 reranked  | 47.57 | 51.29 | 55.38 | 56.96 | 58.45 |
>
> __Q2/Q6: Noise Paragraphs__
>
> Noise paragraphs are defined as non-gold paragraphs retrieved by the retriever, ordered by descending retriever score. This approach simulates real-world retrieval scenarios, reflecting the natural distribution of retrieval outputs. We do not manually select noise paragraphs, since we want to focus on realistic, practical use cases.
>
> __Q3: Better retrievers ≠ Better RAG performance__
>
> If a retriever has better recall@k for a specific k, it typically leads to better reader performance at that k. However, for less robust, peak-then-decline models (e.g., LLaMA 2 and 3), we observe that even when ColBERT has consistently higher recall@k than BM25 across all k, the reader F1 score with ColBERT can still be lower than with BM25 at larger k.
>
> This discrepancy arises because the content retrieved beyond the gold paragraphs plays a critical role, particularly at higher k. Neural retrievers like ColBERT may introduce more semantically complex or noisy content, which can overwhelm noise-sensitive readers. In contrast, BM25, as a lexical retriever, often provides simpler or less distracting context, aligning better with these readers’ preferences despite retrieving fewer gold paragraphs overall.
>
> When reranking is applied (see Q1 for results), ColBERT’s outputs are reordered to reduce noise and prioritize relevance. This improves alignment with less robust readers, allowing better retriever performance to translate into better reader performance. With ColBERT-reranked, the reduction in noise and improved relevance ensure that the benefits of higher recall directly enhance reader outcomes.

---

> ### Author Response · Authors · 2024-11-22
> **Response to Reviewer F1ot (Continued)**
>
> __Q4: Behavior Patterns Across Paradigms__
>
> We appreciate the reviewers' suggestion to explore recent Large Language Model paradigms such as OpenAI O1. However, the cost associated with running even our bare-minimum experiments (across three datasets with top-K and top-positive configurations) is prohibitively high, amounting to approximately $1200. Due to this constraint, we were unable to include OpenAI O1 in our original submission. That said, we have conducted additional experiments with GPT-4 during the rebuttal phase and have included the results in our response to provide further insights. We ran GPT-4o with varying k for NQ, and share the F1 scores below. GPT-4o and GPT-3.5 turbo have the same curve in performance. The difference is that 4o is shifted up about 4-5 points consistently across k’s.
>
> | Model   | 1     | 2     | 5     | 10    | 20    |
> |---------|-------|-------|-------|-------|-------|
> | GPT-3.5 | 49.38 | 50.66 | 52.85 | 54.39 | 55.94 |
> | GPT-4o  | 54.07 | 55.45 | 57.16 | 58.54 | 59.86 |
>
> In our use case, implementing Self-RAG in 'retrieve' mode involves a minimum of three LLM calls per retrieved document per datapoint. With a maximum top-K of 50, this translates to ~200 additional LLM calls per datapoint, making it computationally intensive. Given these constraints, we are not able to report these experiments within the rebuttal period. However, we recognize the significance of this investigation and will look into demonstrating Self-RAG on a smaller subset of data in the revised submission to provide further insights.
>
> __Q6: Alternative Evaluation Metrics__
>
> We appreciate your feedback regarding F1’s limitations and have run an LLM-based evaluation on the models and found the reader trends remain the same. We address this in [this comment thread](https://openreview.net/forum?id=KDXj60FpJr&noteId=KrDAYCQgq7). Please refer to it for a comprehensive response, and feel free to follow up with any additional questions.
>
> __Q7: Retrieval Passage Lengths__
>
> We used the fixed retrieval units from standard corpora that the QA datsaets were associated with: paragraphs for Wikipedia (via KILT) and abstracts/titles for PubMed (BioASQ). We used these as is since we needed annotations for gold documents, and the existing annotations are with respect to these retrieval units.
>
> __Q8: Explanation for different reader trends__
>
> Thank you for your feedback regarding providing more discussion about why some models outperform others. We've addressed this in [this general comment thread](https://openreview.net/forum?id=KDXj60FpJr&noteId=ey7JaljKeE). Please refer to it for a comprehensive response, and feel free to follow up with any additional questions.
>
> __Q9: Parametric knowledge__
>
> We appreciate your suggestion to explore knowledge conflicts and integration. While beyond this paper’s scope, this is an intriguing direction requiring a dedicated setup, such as identifying pretrained knowledge and designing artificial documents that challenge it. This area deserves future investigation in its own right.
>
> __Q10: Practical Implementation Recommendations__
>
> To efficiently determine optimal RAG configurations for new datasets or models:
> 1. Use the observed reader patterns:
>
> - For improve-then-plateau readers (e.g., FLAN, GPT-3.5), begin with a larger k since these models benefit from higher recall and are more robust to noisy documents. Try increasing
> - For peak-then-decline readers (e.g., LLAMA models), prioritize smaller k values (<5) to avoid noise sensitivity.
>
> 2. Domain-Specific Adaptations:
>
> - In general domains (e.g., NQ), dense retrievers like ColBERT outperform sparse retrievers like BM25, providing a strong starting point.
> - In specialized domains (e.g., BioASQ), ColBERT and BM25 perform similarly in retrieval metrics (e.g., recall@k). However, the reader gain due to retrieval improvement is more amplified in specialized domains, making it essential to test both retrievers and focus on even small retrieval improvements.
>
> Setting up experiments based on some of the insights and trends we observe could help avoid the need for exhaustive grid searches across all configurations.

---

> > ### Author Response · Authors · 2024-11-25
> > **Response to Reviewer F1ot (Continued)**
> >
> > Dear reviewer, thank you again for your detailed and thoughtful feedback. We have tried to address your points by running more experiments and adding further analysis and clarifications. We were wondering if our responses address your concerns, or if there are any remaining points where further clarification would be helpful. We’d be happy to provide additional insights if needed. Thank you for your time and thoughtful consideration!

---

> ### Comment · Area_Chair_9EME · 2024-11-26
> **Reminder: Rebuttal Deadline for ICLR 2025**
>
> Dear Reviewer F1ot,
>
> As the rebuttal deadline approaches, please kindly check the papers' discussion threads and respond to the authors' rebuttals. If you haven't had a chance to respond yet, I’d greatly appreciate your input soon. Your insights are invaluable to the authors and the review process.
>
> Thank you for your effort and support!
>
> Best regards,
>
> Area chair

---

> ### Comment · Reviewer_F1ot · 2024-11-27
>
> Having carefully reviewed the author's response, I maintain my original scores. As this is primarily an analysis paper rather than a methodological contribution, the new results on a single dataset/model do not substantially enhance the work's significance. Besides, the revised PDF (not uploaded yet) does not adequately show the additional results and analysis. Furthermore, I think the overall innovation, completeness, and analytical depth are moderate.

---

> > ### Author Response · Authors · 2024-11-28
> > **Response to Reviewer F1ot**
> >
> > Thank you for taking the time to review our response and for carefully considering our additional experiments and clarifications. We appreciate your feedback. Regarding your points:
> >
> > __New results:__
> > >the new results on a single dataset/model do not substantially enhance the work's significance.
> >
> > While we understand results on a single dataset may not sufficiently enhance the paper's significance, we aimed to prioritize feasibility during the rebuttal period and are working to evaluate the RAG pipeline on the rest of the datasets for the final version. We hoped that our additional reranking results (over 2 models) and LLM-based evaluation (over 6 models) would provide insights into the generalizability of our claims. We understand the importance of demonstrating these findings more comprehensively and are working to incorporate broader experimental results in the final version.
> >
> > __Revised Paper:__
> >
> > We have just uploaded the revised PDF to reflect the updated results and analysis, where key changes are in red.
> >
> > __Innovation and Analytical Depth:__
> >
> > While we acknowledge that this paper does not propose innovative evaluation metrics, its primary value lies in systematically and comprehensively analyzing retriever-reader dynamics and robustness. Our goal is to provide a comprehensive evaluation framework and actionable insights that practitioners can directly apply to RAG systems across diverse settings. That said, we do introduce certain novel evaluation angles, such as the slice analysis of RAG performance when signal is present versus absent (Sec. 6), which, to the best of our knowledge, has not been explored in previous work.
> >
> > Regarding analytical depth, we appreciate your suggestions and questions in your review, particularly regarding explanations for reader trends and "better retriever ≠ better RAG performance".  To address the first point, we expanded our analysis to move beyond simply identifying reader trends, instead exploring potential reasons why these trends occur based on model architecture and training details. For the second point, while we previously only described that there exist scenarios where better retriever does not lead to better RAG performance, we now characterize the specific scenarios where better retriever performance does not lead to better reader outcomes and provide potential explanations for why this behavior arises in certain readers. These refined and nuanced analyses have been incorporated into the revised PDF, and we hope they address some of your concerns.
> >
> > Once again, we sincerely thank you for your constructive feedback. It has been helpful in identifying areas for improvement.

---

> > > ### Author Response · Authors · 2024-12-02
> > > **Response to Reviewer F1ot**
> > >
> > > __New results (continued):__
> > > > the new results on a single dataset/model do not substantially enhance the work's significance.
> > >
> > > We understand your concern that results on a single dataset may not substantially enhance the paper's significance. We initially only demonstrated on one dataset (i.e. NQ) due to the time constraint. However, with the recent extension of the rebuttal period, we have run experiments on additional datasets to strengthen the generalizability and significance of the work. We share the new results below:
> > >
> > > -  For the LLM evaluation results: We have added the HotpotQA and BioASQ results to [the original thread about LLM evaluation](https://openreview.net/forum?id=KDXj60FpJr&noteId=KrDAYCQgq7), and find that __the reader trends remain the same.__
> > >
> > > - For the reranker results: Previously, we have demonstrated it only on NQ in this [response](https://openreview.net/forum?id=KDXj60FpJr&noteId=7qBS5tURW0), but have now added the results and analysis for BioASQ below.
> > >
> > > With open-domain (NQ), reranking achieves an average of 2.3 point increase in recall@k across k = 1 to 20, leading to improved reader performance for both improve-then-plateau models like FLAN and peak-then-decline models like LLaMA.
> > >
> > > In contrast, for special-domain (BioASQ), reranking provides only a 0.3 point average recall@k improvement across k = 1 to 20. The resulting reader performance shows minimal gains for FLAN T5 and even declines for LLaMA.
> > >
> > > In specialized domains, precise token-level matching (as used in ColBERT) is important due to the technical nature of queries. The BGE reranker focuses more on broader semantic relationships, which can inadvertently prioritize tangentially relevant documents, amplifying noise and potentially harming performance for noise-sensitive readers like LLaMA.
> > >
> > > __BioASQ results__
> > >
> > >
> > > | Model              | 1     | 2     | 5     | 10    | 20    |
> > > |--------------------|-------|-------|-------|-------|-------|
> > > | LLaMA2 7B          | 32.52 | 32.09 | 30.51 | 28.74 | 27.37 |
> > > | LLaMA2 7B Reranked | 32.67 | 31.08 | 28.58 | 25.28 | 23.23 |
> > >
> > > | Model          | 1     | 2     | 5     | 10    | 20    |
> > > |----------------|-------|-------|-------|-------|-------|
> > > | FLAN T5         | 39.63 | 36.84 | 37.46 | 38.83 | 39.80 |
> > > | FLAN T5 Reranked| 39.09 | 36.43 | 38.56 | 39.70 | 40.71 |

---

### Official Review · Reviewer_mKTy · 2024-11-04

**Soundness:** 2
**Presentation:** 4
**Contribution:** 2
**Rating:** 3
**Confidence:** 4

**Summary:**

The authors introduce an analysis framework, RAGGED, for understanding RAG best practices, namely, whether RAG helps, how many passages to retrieve, which retrievers and reader models to choose, and how that varies across tasks. They conduct this analysis across the retrievers BM25, ColBERT, and Contriever and assess several reader models, evaluating these on NQ (general domain), HotPotQA (complex questions), and BioASQ (specialized domain). Overall, they find that readers vary (some benefit from higher recall at larger retrieval depths; others are sensitive to noise) and that improved retrieval delivers more noticeable gains in specialized domains.

**Strengths:**

1) The paper is very well-written and structured, and hence easy to follow. The research questions posed are clear and the findings are stated and discussed in a well-presented manner.
2) The work empirically identifies trends that are of interest to the community, serving as evidence, for instance, that RAG helps more in specialized domains and that improving the components of a RAG system separately does not guarantee higher end-to-end quality.
3) Thoughtful analysis of RAG systems, which have very quickly become ubiquitous, are very much needed.

**Weaknesses:**

While I admire a thoughtful analysis paper, the present work simply does not explore rich enough questions to balance the lack of novel insight nor does it have enough empirical support for the nuanced claims it tries to extract. To illustrate, the primary non-trivial aspect of the present study are interesting claims like "improved retrieval helps more in specialized domains". This is analytically self-evident (i.e., is not necessarily a new claim) but good empirical support for such claims is always valuable. Unfortunately, the work is essentially trying to support this post-hoc from results on NQ vs. BioASQ, two of the most standard, over-used, and overly-simplistic datasets. To be clear, NQ and BioASQ are still valid and interesting datasets, but the lack of novelty coupled with the factoid and short answer nature and small variety of datasets selected makes it very hard to position this work at ICLR in my current read.

To be very clear, I think work like this is necessary, but to publish this as a long research paper the bar to clear might involve something llike (i) innovating dramatically more on the evaluation front, e.g. assessing human evaluators against a large number of automatic metrics, not just F1 overlap, over a large set of (perhaps contributed) new evaluation benchmarks that vary certain axes more directly or (ii) testing a set of less overused claims and hypotheses, e.g., novel methods that counteract the challenges the authors have observed. Right now, the contribution is simply too small and too ephemeral of an addition to the literature in this (overcrowded) domain.

I encourage the authors to continue this thoughtful analysis line of work and to consider how downstream RAG applications look like in practice --- or alternatively to take some of their statements presented as findings (e.g., improving retrieval doesn't always improve RAG quality) and present deeper insight or a method that contributes to better understanding or better quality.

**Questions:**

N/A

---

> ### Author Response · Authors · 2024-11-22
> **Response to Reviewer mKTy**
>
> Thank you for your encouraging feedback and for appreciating the clarity and structure of our work. We appreciate your recognition of the importance of our findings, particularly in specialized domains and the nuanced retriever-reader interactions.
>
> __W1. Novelty of claims__
>
> Thank you for your question regarding the novelty of the findings. We've addressed this in [this comment thread](https://openreview.net/forum?id=KDXj60FpJr&noteId=w3Xrhj82h3). Please refer to it for a comprehensive response, and feel free to follow up with any additional questions.
>
> __W2. Dataset selection and variety__
>
> We appreciate your feedback and agree that NQ and BioASQ are widely used datasets. That is one reason we chose them since we want to ensure comparability with prior work and alignment with established research practices in the community. We also selected them for their complementarity in terms of domain and task type. Specifically:
> NQ was chosen to represent single-hop open-domain QA.
> BioASQ was included for its specialized-domain nature, providing a contrast to NQ.
> HotPotQA was added to evaluate multi-hop reasoning as a comparison to NQ.
> We would be happy to discuss any other datasets that the reviewer thinks are more suitable for verifying our claims.
>
> __W3. Exploring Alternative Metrics__
>
> We appreciate your feedback regarding F1’s limitations and have run an LLM-based evaluation on the models and found the reader trends remain the same. We address this in [this comment thread](https://openreview.net/forum?id=KDXj60FpJr&noteId=KrDAYCQgq7). Please refer to it for a comprehensive response, and feel free to follow up with any additional questions.
>
> __W4. Depth of Analysis__
>
>  Thank you for your feedback. We agree that deeper insights could enhance the impact of our work, and we’ve been reflecting on areas where we could expand our analysis. We have explored a more in-depth analysis of reader trends based on model details in [this comment thread](https://openreview.net/forum?id=KDXj60FpJr&noteId=ey7JaljKeE). Please refer to it for a comprehensive response, and feel free to follow up with any additional questions. We plan to include this in the revised paper. If there are specific dimensions or insights you’d like us to expand upon, we’d be happy to address your concerns.

---

> > ### Comment · Reviewer_mKTy · 2024-11-25
> >
> > Thanks for the comment. I will re-iterate that I admire a thoughtful analysis paper like this one, and I appreciate that you decided to explore this direction --- I think there's a ton of value in doing this in a more expansive way. Please push more in this front; I do hope that my review above does offer meaningful suggestions on what to do in this respect.
> >
> > However, your response does not address my main concerns, which I quote below:
> >
> > > the present work simply does not explore rich enough questions to balance the lack of novel insight nor does it have enough empirical support for the nuanced claims it tries to extract. [....] the work is essentially trying to support this post-hoc from results on NQ vs. BioASQ, two of the most standard, over-used, and overly-simplistic datasets. [...] to publish this as a long research paper the bar to clear might involve something llike (i) innovating dramatically more on the evaluation front, e.g. assessing human evaluators against a large number of automatic metrics, not just F1 overlap, over a large set of (perhaps contributed) new evaluation benchmarks that vary certain axes more directly or (ii) testing a set of less overused claims and hypotheses, e.g., novel methods that counteract the challenges the authors have observed. Right now, the contribution is simply too small and too ephemeral of an addition to the literature in this (overcrowded) domain.
> >
> > The authors respond:
> >
> > > we chose them since we want to ensure comparability with prior work and alignment with established research practices
> >
> > It is fine to use those _in addition to_ innovating on the (unfortunately very stale) front of benchmarking in this area. The staleness is *not* caused by the authors, but as an analysis paper (whose contribution is not in modeling but in the realm of evaluation) it is crucial to revisit assumptions the average modeling paper tends to make about evaluation datasets.

---

> ### Author Response · Authors · 2024-11-25
> **Response to Reviewer mKTy**
>
> Thank you for your detailed feedback and for acknowledging the value of our work. We deeply appreciate your encouragement to further expand this line of research and your thoughtful suggestions.
>
>
> __W1. Novelty of Claims (continued)__
> > (ii) testing a set of less overused claims and hypotheses
>
> We appreciate your concern about the novelty of our claims and hypotheses. While some of our findings may initially appear intuitive, they address key gaps and unresolved questions in the literature. For example:
> - The observation that RAG configurations can underperform no-context generation has not, to our knowledge, been systematically explored before. This finding underscores the importance of configuration, which is often overlooked in discussions about RAG systems.
> - Similarly, the robustness differences between improve-then-plateau and peak-then-decline models resolve conflicting trends reported in earlier studies (e.g., saturation vs. degradation with increasing k). Our work systematically examines these dynamics across multiple datasets, retrievers, and readers, providing a more nuanced understanding of how noise impacts performance.
> - We also extend prior work on retriever-reader dynamics by demonstrating how domain-specific tasks (e.g., BioASQ) benefit disproportionately from even small retrieval improvements compared to open-domain tasks.
>
> If you know of any prior works that already establish these findings, please share them with us so we can position our contributions more effectively and clarify what remains unique about our analysis.
>
> __W2. Dataset Selection (continued)__
>
> While our current datasets align with community standards and serve as effective starting points for our analysis, we recognize the importance of revisiting evaluation assumptions in an analysis-focused paper. At the time of the dataset selection, we have considered many datasets with emphasis on aspects like how the dataset should include 1) a large corpus to retrieve from to test retriever effectiveness, 2) annotation of gold corpus documents for each question, and 3) free-form answers instead of only MCQ/boolean to test the readers' ability to synthesize coherent, contextually-appropriate answers.
>
> Of the datasets we have considered, here are some that are newer but did not fit our aforementioned selection criteria:
> - StrategyQA [1] emphasizes implicit, multi-step reasoning, similar to HotpotQA, but mainly focuses on boolean questions, not free-text, limiting insights on reader synthesis.
> - QASC [2] also emphasizes multi-hop reasoning, but the corpus is much smaller, which reduces its retrieval complexity: it uses a 17M curated sentences v. the 111M Wiki corpus for NQ, HotpotQA and the 58M corpus for BioASQ. In addition, the answer format is MCQ.
> - PubmedQA [3] focuses on yes/no/maybe questions using predefined PubMed abstracts as context. However, they do not have a retrieval aspect since the relevant context is already provided with the context.
>
>
> __W3. Exploring Alternative Metrics (continued)__
>
> >  (i) innovating dramatically more on the evaluation front, e.g. assessing human evaluators against a large number of automatic metrics, not just F1 overlap, over a large set of (perhaps contributed) new evaluation benchmarks that vary certain axes more directly
>
> We incorporated LLM-based evaluations using PROMETHEUS-7B, which achieves a Pearson correlation of 0.897 with human evaluators [4], which makes it a reasonable alternative to costly and time-intensive human evaluation. Using PROMETHEUS, we validated that our reader trends (e.g., improve-then-plateau vs. peak-then-decline) remain consistent. This demonstrates that our findings are robust across evaluation methods.
>
> If there are additional evaluation methodologies you feel are necessary to validate our findings, we welcome your suggestions and would be happy to consider them for future work.
>
> [1] Geva, Mor, et al. "Did aristotle use a laptop? a question answering benchmark with implicit reasoning strategies." Transactions of the Association for Computational Linguistics 9 (2021): 346-361.
>
> [2] Khot, Tushar, et al. "Qasc: A dataset for question answering via sentence composition." Proceedings of the AAAI Conference on Artificial Intelligence. Vol. 34. No. 05. 2020.
>
> [3] Jin, Qiao, et al. "Pubmedqa: A dataset for biomedical research question answering." arXiv preprint arXiv:1909.06146 (2019).
>
> [4] Kim, Seungone, et al. "Prometheus 2: An open source language model specialized in evaluating other language models." arXiv preprint arXiv:2405.01535 (2024).

---

### Official Review · Reviewer_L2xh · 2024-11-11

**Soundness:** 4
**Presentation:** 4
**Contribution:** 3
**Rating:** 8
**Confidence:** 4

**Summary:**

This paper studies how the components of a retrieval-augmented generation (RAG) system impact its performance on a variety of tasks. They propose an evaluation framework called RAGGED that  evaluates the impact of more context, inclusion of irrelevant information, and the interaction between task type model performance. They compare combinations of three retrievers (BM25, ColBERT, Contriever) and four readers (FLAN, LLaMa, GPT, Claude) on multiple datasets and present a series of findings that make the case for designing RAG frameworks that are task-specific and tailored to their use case.

**Strengths:**

- With increasing adoption of RAG for a variety of tasks, this is a useful direction of research that helps researchers understand when and how to use this set up. These findings could be potentially useful for deciding how to configure a RAG framework based on the task.

- The RAGGED framework proposed in this paper is well-designed and addresses critical questions at each step of the pipeline. The authors aim to release the code for this framework which would enable easier replication and extension to include more models/experiments.

- The paper is well-written and the results are presented in a very easy to understand manner. I really appreciate how Table 1 and the figures that follow communicate the core findings succinctly. The appendix also provides useful additional details for the experiments.

**Weaknesses:**

- The paper does not compare more state of the art closed-source models like GPT-4 and Claude 3 Opus. It’s understandable that cost may be a hindrance, but it would be helpful to conduct even a smaller set of experiments using stronger models so the takeaways from this paper are more broadly applicable.

- Unigram F1 doesn’t strike me as the ideal metric for evaluating reasoning quality in LLMs. Perhaps the authors could have explored semantic embedding-based approaches or by simply asking another LLM to verify the (output, label) pair for accuracy.

- While the takeaways are interesting, it would have been helpful to hear the authors’ thoughts on why certain models outperform others in different settings.

**Questions:**

- Did the authors try any kind of prompt engineering to see if it improves performance?
- The FanOutQA benchmark (Zhu et al., ACL 2024) strikes me as a particularly suitable Wikipedia-based testbed for this framework. Did the authors attempt to include it in their evaluation?
- Another increasingly important factor in RAG systems in multilinguality, which impacts both retrievers and readers. It would be useful to include at least one multilingual dataset, like NoMIRACL (Thakur et al., EMNLP 2024), in this testing framework.

---

> ### Author Response · Authors · 2024-11-22
> **Response to Reviewer L2xh**
>
> Thank you for your thoughtful review and for highlighting the utility of our work in guiding RAG configurations. We appreciate your recognition of the well-designed RAGGED framework, the clarity of our results (e.g., Table 1 and figures), and our commitment to open-sourcing the framework for broader use.
>
> __W1. Evaluating with new closed source models -  GPT-4 and Claude 3 Opus__
>
> Thanks for the suggestion! Per your suggestion,we ran GPT-4o with varying k for NQ, and share the F1 scores below. GPT-4o and GPT-3.5 turbo have the same curve in performance. The difference is that 4o is shifted up about 4-5 points consistently across k’s.
> | Model   | 1     | 2     | 5     | 10    | 20    |
> |---------|-------|-------|-------|-------|-------|
> | GPT-3.5 | 49.38 | 50.66 | 52.85 | 54.39 | 55.94 |
> | GPT-4o  | 54.07 | 55.45 | 57.16 | 58.54 | 59.86 |
>
> Claude Opus is significantly more expensive than GPT-4o and would cost $1300 to include. Due to this constraint, we were unable to include this model in our original submission.
>
> __W2. Unigram F1 as the metric__
>
> We appreciate your feedback regarding F1’s limitations and have run an LLM-based evaluation on the models and found the reader trends remain the same. We address this in [this comment thread](https://openreview.net/forum?id=KDXj60FpJr&noteId=KrDAYCQgq7). Please refer to it for a comprehensive response, and feel free to follow up with any additional questions.
>
>
> __W3. Why do some models outperform others__
>
> There are two primary types of readers observed in our experiments:
> Peak-then-Decline Behavior: Certain models such as LLAMA and Claude show sensitivity to noisy documents, leading to performance degradation as the number of retrieved passages (k) increases beyond a certain point.
> Improve-then-Plateau Behavior: Certain models such as GPT and FLAN are more robust to noise, continuing to benefit from additional context until performance plateaus.
> Since we don’t have access to the details of the closed source models, we can provide hypotheses according to the open-source model (LLaMa belonging to the peak-then-decline behavior and the Flan models belonging to the improve-then-plateau family).
>
> FLAN, an improve than plateau model family, incorporates additional strategies explicitly designed to handle noisy or diverse contexts. It employs denoising strategies, such as a mixture-of-denoisers, during training to improve its robustness to irrelevant or noisy context. These enhancements enable it to filter out noise more effectively. On the other hand, LLaMa’s training predominantly relies on next-token prediction with limited exposure to noisy or retrieval-specific scenarios, making it sensitive to noise at higher k.
>
>
> __Q1. Prompt engineering__
>
> Thank you for the suggestion. We will try chain-of-thought and update with the results.
> We appreciate the reviewers' suggestion to explore the impact of prompt engineering variations such as Chain-of-Thought (CoT) and Self-Ask. Based on prior research [2], CoT prompting demonstrates performance gains primarily with large models (∼100B parameters), while smaller models often fail to benefit. In our experiments with various open-source models (Llama2-7B, Llama2-70B, Flan-T5, and Flan-UL2), CoT prompting did not yield consistent reasoning or structured responses, particularly with smaller LLMs. Despite trying different CoT prompts and response configurations, the models struggled to articulate reasoning steps or follow instructed formats.
>
> __Q2. The FanOutQA benchmark__
>
> Thank you for your suggestion and are looking into adding this to the final version. We were not aware of FanOutQA at the time of the paper as FanOutQA was only published a month prior to the submission. We are looking into adding this dataset, but will not able to update with new results within the rebuttal period since there is a large set of experiments to cover (i.e. all combinations of 8 models, 2 retrievers, 3 retrieval models (top-k, top-positive, no-context), and a range of k's). We highly appreciate your dataset suggestion and appreciate how it can enrich our analysis.
>
> __Q3. Multilingual QA__
>
> Multilinguality is an important aspect, though our current study focuses on RAG configuration in Englisht to understand foundational patterns. Extending the framework to multilingual settings introduces other factors such as language-specific retrieval which deserves its own dedicated exploration.
>
> [1] Kim, Seungone, et al. "Prometheus 2: An open source language model specialized in evaluating other language models." arXiv preprint arXiv:2405.01535 (2024).
>
> [2] Wei, Jason, et al. "Chain-of-thought prompting elicits reasoning in large language models." Advances in neural information processing systems 35 (2022): 24824-24837.

---

> > ### Author Response · Authors · 2024-11-25
> > **Response to Reviewer L2xh**
> >
> > Dear reviewer, thank you again for your detailed and thoughtful feedback. We have tried to address your points by running more experiments and adding further analysis. We were wondering if our responses sufficiently address your concerns, or if there are any remaining points where further clarification would be helpful. We’d be happy to provide additional insights if needed. Thank you for your time and thoughtful consideration!

---

> > ### Comment · Reviewer_L2xh · 2024-11-26
> > **Response to Authors**
> >
> > Thank you for providing these clarifications and conducting the additional experiments, they are quite helpful. I look forward to reading the final version and have updated my scores accordingly.

---

> > > ### Author Response · Authors · 2024-11-26
> > > **Response to Reviewer L2xh**
> > >
> > > Thank you for your thoughtful feedback and for taking the time to update your scores! We are glad that the clarifications and additional experiments were helpful. We look forward to incorporating these improvements into the final version.

---

> ### Comment · Area_Chair_9EME · 2024-11-26
> **Reminder: Rebuttal Deadline for ICLR 2025**
>
> Dear Reviewer L2xh,
>
> As the rebuttal deadline approaches, please kindly check the papers' discussion threads and respond to the authors' rebuttals. If you haven't had a chance to respond yet, I’d greatly appreciate your input soon. Your insights are invaluable to the authors and the review process.
>
> Thank you for your effort and support!
>
> Best regards,
>
> Area chair

---

### Author Response · Authors · 2024-11-22
**Response to Reviewer Comments: Clarifying Novelty in Our Approach**

While we understand some findings may seem intuitive, it is not necessarily the case that they hold across all models and question types. As such, phenomena that are not attested to in the literature should be examined in a systematic fashion.

Below are key findings from our study and how they provide a more nuanced and comprehensive view compared to existing literature. These comparisons are cited in the paper, but we will revise the text to make these distinctions clearer:

- __RAG vs. No-Context Generation__:

While [1] demonstrate that retrieval-augmented generation (RAG) outperforms closed-book (no-context) generation and achieved state-of-the-art results for QA tasks at the time, we extend this understanding. Specifically, we show that with the wrong configuration, RAG readers can perform worse than no-context generation (Sec. 4). For some readers, benefitting from RAG requires a large enough k while others require a small enough k. This highlights the importance of configuration, a point largely unexplored in [1].

*To the best of our knowledge, we are the first to study the comparison with no-context performance more comprehensively across RAG configurations and question complexities (hops)/domains.*

- __Reader trends across k__:

Prior works offer limited experiments and seemingly conflicting insights into how readers perform with increasing k:
[2] observed saturation in reader performance as k increases, but their analysis was limited to specific question types and retrievers.
[3,4] reported performance degradation at higher k, without fully explaining the underlying causes.
Our study resolves these inconsistencies by conducting more comprehensive experiments that span multiple datasets, retrievers, and readers. This allows us to identify reader robustness to noise as the key determinant of trends across k.

*To the best of our knowledge, we are the first to conduct a granular slice analysis of robustness by examining the impact of noise on different reader models specifically in the presence and absence of signal (Sec 6), resulting in key findings such as: RAG can perform worse than no-context even in the presence of sufficient signal*.

- __Retriever Importance Across Domains and Tasks__:

While [4] found a positive correlation between retriever and reader performance on a single dataset, we present a more nuanced perspective *across question types*:
Domain-specific tasks (e.g., biomedical QA) gain significantly more from improved retrievers than open-domain tasks.
Single-hop questions benefit more than multi-hop, though the gap is smaller than the gap between special domain v. open domain.
These findings provide a broader, more detailed understanding of how retriever performance impacts reader effectiveness across different scenarios.

Nonetheless, If you find similar findings in other papers, we are happy to incorporate and discuss these papers.

[1] Patrick Lewis, Ethan Perez, Aleksandra Piktus, Fabio Petroni, Vladimir Karpukhin, Naman Goyal,
Heinrich Küttler, Mike Lewis, Wen-tau Yih, Tim Rocktäschel, et al. Retrieval-augmented genera-
tion for knowledge-intensive nlp tasks. Advances in Neural Information Processing Systems, 33:
9459–9474, 2020.

[2] Nelson F. Liu, Kevin Lin, John Hewitt, Ashwin Paranjape, Michele Bevilacqua, Fabio Petroni, and
Percy Liang. Lost in the middle: How language models use long contexts, 2023.

[3] Florin Cuconasu, Giovanni Trappolini, Federico Siciliano, Simone Filice, Cesare Campagnano,
Yoelle Maarek, Nicola Tonellotto, and Fabrizio Silvestri. The power of noise: Redefining retrieval
for rag systems. In Proceedings of the 47th International ACM SIGIR Conference on Research
and Development in Information Retrieval, pp. 719–729, 2024.

[4] Paulo Finardi, Leonardo Avila, Rodrigo Castaldoni, Pedro Gengo, Celio Larcher, Marcos Piau, Pablo
Costa, and Vinicius Caridá. The chronicles of rag: The retriever, the chunk and the generator.
arXiv preprint arXiv:2401.07883, 2024.

---

### Author Response · Authors · 2024-11-22
**Response to Reviewer Feedback: Using Semantic-Based Metrics for Evaluation**

We appreciate the reviewers' feedback regarding F1’s limitations. While we chose F1 for its simplicity and alignment with prior work, we agree that it may not fully reflect nuanced semantic equivalence. To address this, we ran an LLM-based evaluation of the models for all of the datasets using Prometheus [1], which has a Pearson correlation of 0.897 with human evaluators. We find that __the conclusions about reader trends do not change: the same reader trends apply to the same models__ (LLaMa and Claude still exhibit peak-then-decline while GPT-3.5 and FlanT5 still exhibit improve-then-plateau). We use Prometheus-7b-v2.0 to evaluate the correctness of the generated answer against the gold answer on a 5-point scale, where 1 is the least correct and 5 is the most correct:

Dataset: Natural Questions
| Model          | 1     | 2     | 5     | 10    | 20    |
|----------------|-------|-------|-------|-------|-------|
| LLaMA2 7B      | 3.22  | 3.19  | 3.06  | 2.84  | 2.70  |
| FlanT5         | 3.27  | 3.43  | 3.56  | 3.63  | 3.64  |
| GPT-3.5        | 3.48  | 3.56  | 3.63  | 3.67  | 3.72  |
| Claude Haiku   | 3.19  | 3.16  | 2.92  | 2.68  | 2.36  |

Dataset: HotpotQA
| Model          | 1     | 2     | 5     | 10    | 20    |
|----------------|-------|-------|-------|-------|-------|
| LLaMA2 7B      | 3.20  | 3.14  | 3.03  | 2.93  | 2.87  |
| FlanT5         | 3.12  | 3.11  | 3.17  | 3.19  | 3.21  |
| GPT-3.5        | 3.65  | 3.75  | 3.80  | 3.83  | 3.85  |
| Claude Haiku   | 3.26  | 3.03  | 2.63  | 2.38  | 2.27  |

Dataset: BioASQ

| Model          | 1     | 2     | 5     | 10    | 20    |
|----------------|-------|-------|-------|-------|-------|
| LLaMA2 7B      | 2.94  | 2.97  | 2.89  | 2.82  | 2.63  |
| FlanT5         | 2.94  | 3.03  | 3.14  | 3.19  | 3.20  |
| GPT-3.5        | 3.21  | 3.14  | 3.19  | 3.21  | 3.23  |
| Claude Haiku   | 2.90  | 2.86  | 2.64  | 2.48  | 2.25  |

[1] Kim, Seungone, et al. "Prometheus 2: An open source language model specialized in evaluating other language models." arXiv preprint arXiv:2405.01535 (2024).

---

### Author Response · Authors · 2024-11-22
**Response to Reviewer Comments: Explaining Reader Trends**

There are two primary types of readers observed in our experiments:
- __Peak-then-Decline Behavior__: Models including those from the LLAMA and Claude families show sensitivity to noisy documents, leading to performance degradation as the number of retrieved passages (k) increases beyond a certain point.
- __Improve-then-Plateau Behavior__: Models including those from the GPT and FLAN families are more robust to noise, continuing to benefit from additional context until performance plateaus.

Since we don’t have access to the details of the closed-source models, we will focus on providing hypotheses according to the open-source model (LLaMa belonging to the peak-then-decline behavior and the FLAN models belonging to the improve-then-plateau family).

On the one hand, FLAN, an improve-then-plateau model family, incorporates additional strategies explicitly designed to handle noisy or diverse contexts. It employs denoising strategies, such as a mixture-of-denoisers, during training to improve its robustness to irrelevant or noisy contexts. These enhancements enable it to filter out noise more effectively.

On the other hand, LLaMa’s training predominantly relies on next-token prediction with limited exposure to noisy or retrieval-specific scenarios, making it sensitive to noise at higher k.

We also note that there are some model architecture features that *alone do not determine reader behavior*:
- __Context window size__: Models with longer context limits like LLaMa2 (4k tokens) don't necessarily process a larger number of contexts better than models with smaller context limits like FLAN (2k tokens).

- __Encoder-decoder v. decoder__: LLaMa is a decoder-only model that displays peak-then-decline behavior, but GPT models are also decoder-only and instead display improve-then plateau behavior.

---

### Meta-Review · Area_Chair_9EME · 2024-12-22

**Metareview:**

Summary of the paper: This paper introduces RAGGED, an evaluation framework designed to analyze the performance of RAG systems. It investigates how various components—such as different retrievers (BM25, ColBERT, Contriever) and reader models (FLAN, LLaMa, GPT, Claude)—impact RAG effectiveness across diverse tasks and datasets, including NQ, HotPotQA, and BioASQ. Key findings reveal that the performance of reader models varies significantly based on retrieval depth and sensitivity to irrelevant information, highlighting the importance of task-specific RAG configurations. The framework aims to provide insights for optimizing RAG system deployment by systematically evaluating component interactions and performance characteristics in both general and specialized domains.

Strengths of the paper:
- Clear Presentation: In general, this paper is well-written and structured. Clear tables and figures effectively communicate core insights, making the results easy to understand.
- Resource Accessibility: The implementation of the proposed framework is publicly available, along with detailed documentation of experimental setups, promoting further research and application in the field.
- Systematic Analysis: The study employs a comprehensive approach to RAG configuration, systematically examining multiple dimensions, models, and interactions, providing valuable empirical evidence about component relationships and particularly regarding the interplay between retriever and reader performance, contributing to a deeper understanding of their dynamics.

Weaknesses of the paper:
- Lack of Comparison with Advanced LLMs (Reviewer L2xh): This paper does not compare SOTA closed-source LLMs (e.g. GPT-4), limiting the applicability of its findings and insights.
- Underexplored Retrieval Methods (Reviewer F1ot): This paper does not incorporate contemporary retrieval techniques, such as BGE and reranking mechanisms, and does not evaluate the impact of modern search engines.
- Inadequate Evaluation Metrics (Reviewer L2xh): The use of Unigram F1 as a metric for reasoning quality is insufficient; exploring semantic embedding-based metrics or using another LLM for verification could enhance evaluation.
- Insufficient Empirical Support (Reviewer mKTy, F1ot, e7Uq): Claims such as "improved retrieval helps more in specialized domains" lack robust empirical evidence, relying on standard datasets (NQ, BioASQ, HotpotQA) that may not provide novel insights or generalizability to broader contexts.
- Limited Depth of Analysis (Reviewer mKTy, F1ot, e7Uq): The current analysis of the paper does not delve deeply enough into significant questions. It lacks sufficient novelty and empirical support necessary to substantiate its nuanced claims.

Reasons for the decision: After considering the rebuttal, I acknowledge that some of the concerns raised by the reviewers have been addressed, as noted in the additional comments from the reviewer discussion. However, two significant issues remain unresolved: insufficient empirical support and limited depth of analysis. Having reviewed the authors' responses, I understand their rationale for including only NQ, BioASQ, and the newly added HotpotQA, which stems from the available annotations in existing datasets. Following discussions with reviewers mKTy and F1ot during the reviewer discussion period, I find myself more aligned with their perspective. Currently, the depth of analysis provided by the paper does not meet the standards for publication as a long research paper at ICLR. An analysis paper at ICLR should provide insights that are valuable not just in the immediate future, but also for years to come. In light of these considerations, I lean toward rejection.

**Additional Comments On Reviewer Discussion:**

I appreciate the efforts made by the authors during the rebuttal. The authors addressed part of the concerns below by adding many experiments:
Weaknesses of the paper:
- Lack of Comparison with Advanced LLMs (Reviewer L2xh): The authors include results of GPT4o.
- Underexplored Retrieval Methods (Reviewer F1ot): The authors ran experiments using bge-reranker-large.
- Inadequate Evaluation Metrics (Reviewer L2xh): The authors ran an LLM-based evaluation of the models for all of the datasets using Prometheus.
However, the other two major concerns remain (insufficient empirical support and limited depth of analysis).

---

### Decision · Program_Chairs · 2025-01-22

Reject